# Longitudinal proteomic profiling of cerebrospinal fluid in untreated multiple sclerosis defines evolving disease biology

Peter Kosa[1], Shinji Ashida[1], Keith Lumbard[2], Jing Wang[2], C. Jason Liang[3], Ruturaj Masvekar[1], Yujin Kim [1], Mihael Varosanec[1], Lori Jennings [4] & Bibiana Bielekova [1] ✉

Multiple sclerosis (MS) is a chronic inflammatory disease of the central nervous system, but the molecular mechanisms underlying its course remain incompletely understood. We measured 4789 cerebrospinal fluid proteins in 1040 samples from 438 individuals with MS and controls followed longitudinally. To isolate disease-related biology, we adjusted for normal aging, sex, while also measuring residual effects of demographic and genetic covariates. Here we show that 3714 proteins are significantly associated with twelve clinical and imaging outcomes, highlighting processes linked to viral infection, disruption of epithelial barriers, stromal cell–mediated tissue remodeling, demyelination, and synaptic and neuronal loss. We also find strong sex-related differences: men show greater activation of pathways associated with tissue injury and disability accumulation, whereas women upregulate neurodevelopmental programs that may promote resilience or repair. These molecular maps of MS natural history provide a framework for understanding disease mechanisms and a resource for future drug development.

Chronic diseases evolve over decades. Even when initiated by a single cause, ongoing tissue damage triggers compensatory mechanisms aimed at preserving function. If the cause persists, its continuous interaction with the injured and remodeled tissue creates secondary mechanisms, affected by comorbidities, genetic, demographic, and environmental factors[1–3]. The resulting multiplicity and heterogeneity of disease mechanisms make chronic diseases progressively difficult to treat.

Multiple sclerosis (MS) is such a prototypical chronic disease of the central nervous system (CNS). Current disease-modifying treatments (DMTs) inhibit the hallmark formation of CNS inflammatory-demyelinating lesions by >90%[4]. However, DMT's ability to inhibit disability progression decreases strongly with patients' age[5], consistent with the broadening of pathogenic mechanisms. Postmortem studies identified compartmentalized inflammation, oxidative stress,

mitochondrial dysfunction, excitotoxicity, toxic astrogliosis, endoplasmic reticulum stress, and others in MS CNS[6–12]. However, by analyzing single/few processes in limited CNS tissue once, pathology studies cannot elucidate how disease mechanisms develop, interact, or contribute comparatively to disease progression. While transcriptomic studies measure hundreds of markers, cost and sample scarcity preclude large datasets, limiting their generalizability.

Cerebrospinal fluid (CSF) proteomics lacks spatial information. But when applied to a large cohort of deeply characterized people with MS (pwMS) spanning the entire phenotypic spectrum, CSF proteomics provides uniquely complementary information to pathology and transcriptomic studies.

Thus, we aimed to use CSF biomarkers to define intrathecal molecular mechanisms underlying key aspects of MS evolution and to

[1]Neuroimmunological Diseases Section, Laboratory of Clinical Immunology and Microbiology, National Institute of Allergy and Infectious Diseases, National Institutes of Health, Bethesda, MD, USA. [2]Clinical Monitoring Research Program Directorate, Frederick National Laboratory for Cancer Research, Frederick, MD, USA. [3]Biostatistics Research Branch, National Institute of Allergy and Infectious Diseases, National Institutes of Health, Bethesda, MD, USA. [4]Novartis Institutes for Biomedical Research, Cambridge, MA, USA. ✉e-mail: Bibi.Bielekova@nih.gov

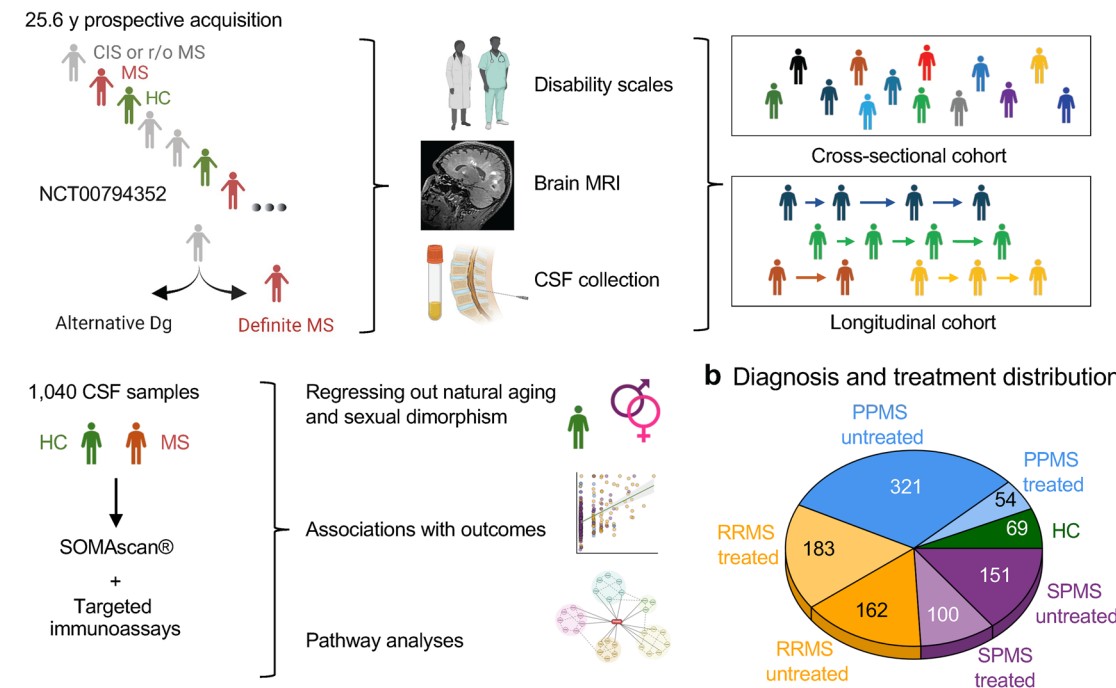

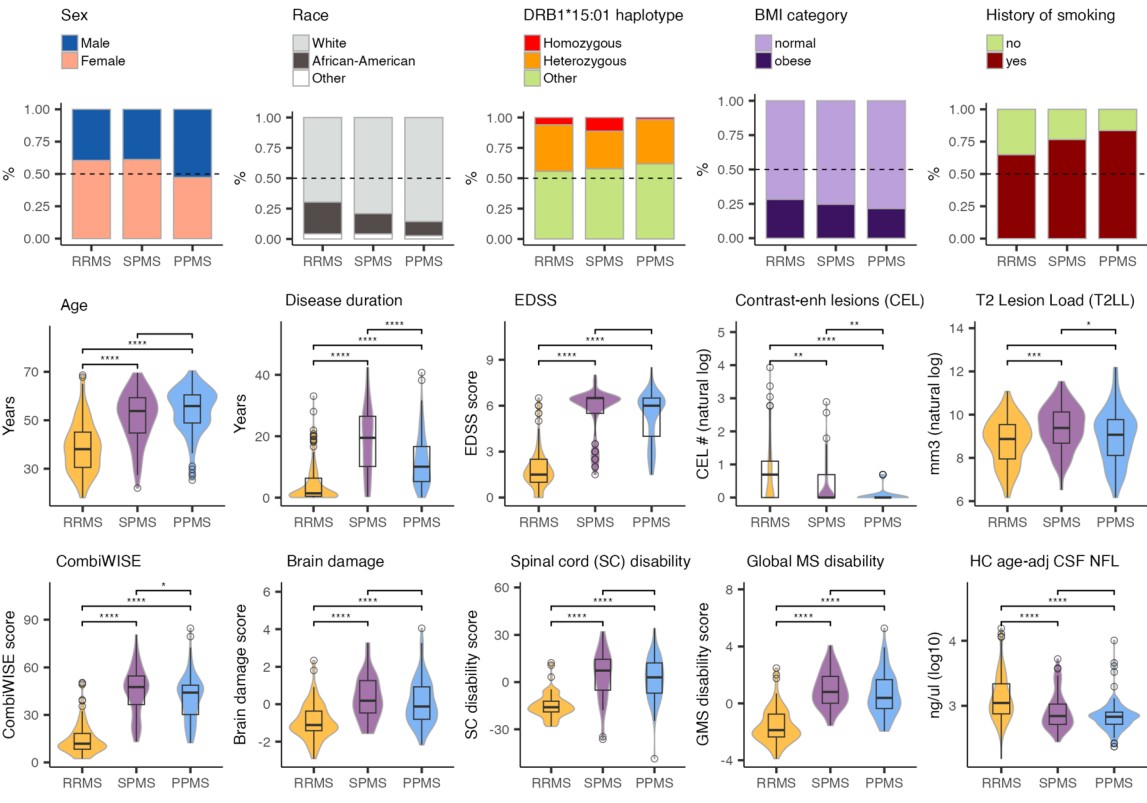

explore how demographic, behavioral, and genetic factors may influence disease course. We identified processes associated with the formation of acute (contrast-enhancing) MS lesions (CELs), their accumulation as T2 lesion load (T2LL), and accrual of cognitive and physical disabilities. In the longitudinal cohort, we show that some proteins/pathways reflect disability progression rates, while others can predict them.

## Results

### Study design

Data were prospectively collected over 25.6 years, database inputted, and quality controlled (QC). The data distributions (Fig. 1) align with population-based MS cohorts[13,14].

3714 human proteins were measured with a high signal-to-noise ratio (SNR; Supplementary data 1) in 1040 CSF samples; 971 samples

**Fig. 1 | Study design and demographic data. a** Patients with suspected or confirmed multiple sclerosis (MS) and healthy controls (HC) were enrolled into the Natural History Protocol (ClinicalTrials.gov: NCT00794352). All participants underwent a neurological exam, paraclinical testing, and brain and spinal cord MRI. CSF was collected at baseline with optional follow-up. Final diagnoses confirmed MS or alternative conditions; HCs underwent the same evaluations. Patients were prospectively recruited over >20 years, yielding both cross-sectional and longitudinal cohorts. In total, 1040 CSF samples were obtained from the MS and HC groups. CSF proteins were profiled with the SOMAscan® assay (4789 epitopes), and additional biomarkers quantified by in-house ELISAs. HC data were used to adjust MS samples for age and sex. Associations of adjusted somamers with clinical, disability, and demographic outcomes were tested by linear regression (sex as covariate). Significant biomarkers were interrogated using Ingenuity Pathway Analysis (IPA) and public databases (STRING, GTEx, Human Protein Atlas) to identify biological processes. **b** Pie chart shows diagnostic and treatment distribution: 69 HC samples (green) and MS subtypes–relapsing-remitting (RRMS, orange), primary progressive (PPMS, blue), and secondary progressive (SPMS, purple). **c** Top row: bar plots show subgroup proportions by sex, race (White, African American, other), DRB1*15:01/03 haplotype (homozygous, heterozygous, other), BMI (normal <30, obese ≥30), and smoking status at first visit. Middle and bottom rows: violin and box plots display demographic and clinical features across RRMS (orange), SPMS (purple), and PPMS (blue) at first visit. Variables include age, disease duration, Expanded Disability Status Scale (EDSS), number of contrast-enhancing lesions (CEL#), T2 lesion load (T2LL), Combinatorial age-adjusted disability scale (CombiWISE), brain damage, spinal cord (SC) disability, global MS (GMS) disability, and HC-adjusted neurofilament light chain (NEFL). Box plots show medians, quartiles, and whiskers (1.5 × interquartile range [IQR]). Statistical significance was assessed by two-sided one-way ANOVA with post-hoc pairwise Wilcoxon rank-sum tests. Created in BioRender. Kosa, P. (2025) https://BioRender.com/bl20ikb.

belonged to 394 MS patients (162 relapsing remitting [RRMS], 151 secondary-progressive [SPMS], and 321 primary-progressive [PPMS] untreated samples; Fig. 1; Supplementary Table 1). We subtracted the effects of natural aging and sex on CSF proteome using 69 CSF samples from 44 healthy controls (HC) (Supplementary data 2).

Unless noted otherwise, results use only data from untreated pwMS. All p-values are adjusted for multiple comparisons, and all results are provided in Supplementary data 3.

## Proteomic differences between pwMS and HC point to intrathecal viral, possibly EBV infection associated with lymphopoiesis and demyelination

1668 protein levels differed between pwMS and HC (Fig. 2). Merging the Human Protein Atlas data with the re-analyses of MS transcriptomics studies (i.e., brain single nuclei (sn)RNAseq[15–19] and CSF single cell (sc)RNAseq[20–25]) identified cellular origin of studied proteins (Fig. 3).

Expectedly, immune cell-specific proteins were preferentially increased in pwMS (Fig. 2a). Plasmablasts/plasma cells (PB/PC) proteins were increased by 90%, followed by monocyte (+56%) and macrophage proteins (+52%). Only oligodendrocyte proteins were reduced in MS by 73% (Supplementary data 4). By identifying PB/PC-centric inflammation and demyelination, CSF proteomics reproduces MS pathology by capturing the entire intrathecal compartment in living pwMS.

Ingenuity Pathway Analysis (IPA) that integrates 39 public databases, >600,000 omics data and >8 million curated scientific observations aggregated MS-specific proteins to 58 pathogenic functions (Fig. 2b, Supplementary data 5-8) and following summary (Fig. 2c): pwMS have upstream activation of anti-viral immunity represented by type I (IFNA2), type II (IFNG) and type III (IFNL1) interferons, IFN-related transcription factors (IRF1, −5, −9, NFKB1), proteins upregulated in virally-infected cells (MAVS, PARP9, DOCK8) and cytokines IL21 (activates B cells) and IL27 (associates with EBV-induced EBI3 to form cytokine with anti-viral activity). Considering the link between EBV and MS[26] and that differentiation of latently infected B cells into PB/PC[27], which are uniquely enriched in MS CNS, reactivates EBV, this summary model is consistent with ongoing intrathecal EBV infection, although it does not prove it.

To examine the evolution of MS-related proteome, we calculated yearly progression slopes for 3714 MS-linked proteins and 182 pathways in 110 pwMS with ≥2 untreated CSF samples collected at least 1.5 years apart (median 4.0 years, range 1.6–16.8). 910 proteins and 36 pathways significantly changed, 79% increasing and 21% decreasing with MS evolution. Based on changing protein slopes, the IPA predicted upstream activation of inflammatory mediators and mesenchymal cell growth factors (Fig. 2d), but also intracellular viral sensors RIGI and STING1 and transcriptional factor IRF7 that regulates EBV latency[28,29]. In contrast, anti-apoptotic MAVS that

mediate anti-viral defense by activating NLRP3 inflammasome decreased with MS evolution. Macromolecularly, growth of lymphoid organs (e.g., tertiary lymphoid follicles; TLF), and migration and activation of vascular endothelial cells (VEC) and fibroblasts, increased with MS evolution.

Thus, CSF biomarkers suggest that untreated MS evolution is characterized by ongoing injury to epithelial barriers and inflammation-associated stromal cell-mediated tissue remodeling (see Supplementary Note 1), which likely facilitates the compartmentalization of inflammation within CNS tissue.

## Proteomics captures the inflammatory/demyelinating nature of CELs with activation of anti-viral immunity and substantial differences between male and female pwMS

Despite subtracting physiological age and sex effects, most MS outcomes-associated proteins showed profound residual differences between male and female pwMS (Fig. 3c). Multiple linear regression models (Supplementary data 3) mapped how sex affects disease mechanisms.

The first feature of MS are CNS lesions visible on magnetic resonance imaging (MRI) as CELs. Of 258 proteins significantly linked to CEL numbers (CEL#), 37% correlated positively (Fig. 4a–c). Proteins previously associated with CEL formation: CXCL13, MMP9, IL12p40[30,31], and neurofilaments[32,33] (NEFH > NEFL)[34–38] exerted high effect sizes (Supplementary data 3).

Most proteins increasing with CEL# were immune cells derived (Fig. 4c) and sex affected. But only B cell- ($p = 0.028$) and macrophage-enriched proteins ($p = 0.0383$) showed likely pathogenic bias (i.e., over-representation among positive correlations), while PB/PC-enriched proteins were biased towards negative CEL# correlations ($p = 0.0415$; Fig. 3). Males had increased levels of most proteins positively correlating with CEL#. Among these were B cell (CD40, FCRL1, and FAIM3), T cells (LAG3, CRTAM, LTB), myeloid cell proteins (TREM1, LYZ, LILRA5, IL1RN), and extracellular matrix (ECM)-degrading enzymes (MMP7, −8, CTBS). Females had increased myeloid markers SPP1 (osteopontin) and CSF1R (Supplementary Fig. 1).

Based on the directionality of proteins' correlations with CEL# IPA predicted upstream activation of B cell (BCR) and T cell (TCR) receptors and related inflammatory molecules (Fig. 4d–f). Anti-viral proteins HAVCR1 (controls T cell accumulation in inflamed CNS[39]), IRF7, MAVS, and DOCK8 were also activated in CEL-associated samples (Fig. 4f). We orthogonally validated IPA-predictions for proteins with CNS-detectable transcripts, such as upregulation of CLEC7A and FCGR2A in active MS lesions (Supplementary data 3).

IPA predicted CEL-linked downmodulation of lipid metabolism regulators PPARG, RXRA/B, and interferon-induced IFITM3 that protects cells from viral entry by affecting cholesterol[40]. PwMS forming CELs also had low concentrations of oligodendrocyte- ($p = 0.028$), astrocyte- ($p = 0.0368$), excitatory neurons (ExNeurons; $p = 0.0368$),

**a**   Differences in cell-specific markers that differentiate MS from HC

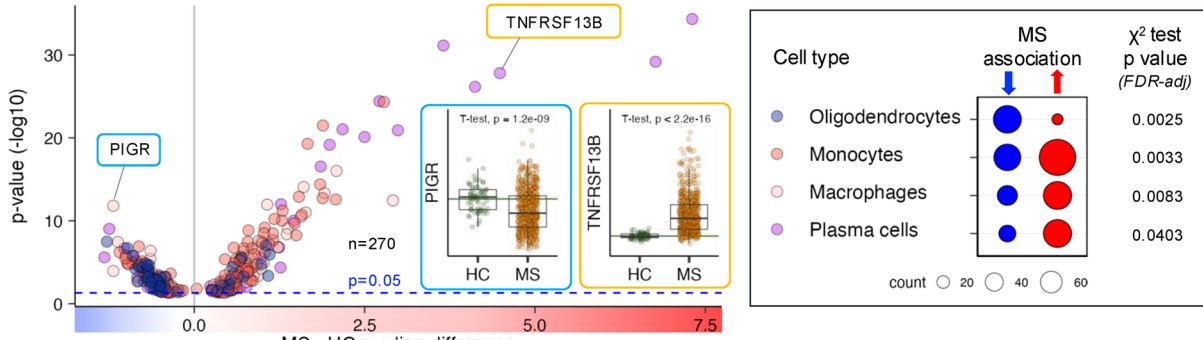

**b**   IPA Diseases & Functions results of proteins differentiating MS from HC

**c**   Graphical summary of IPA analysis of proteins significantly differentiating untreated MS from HC

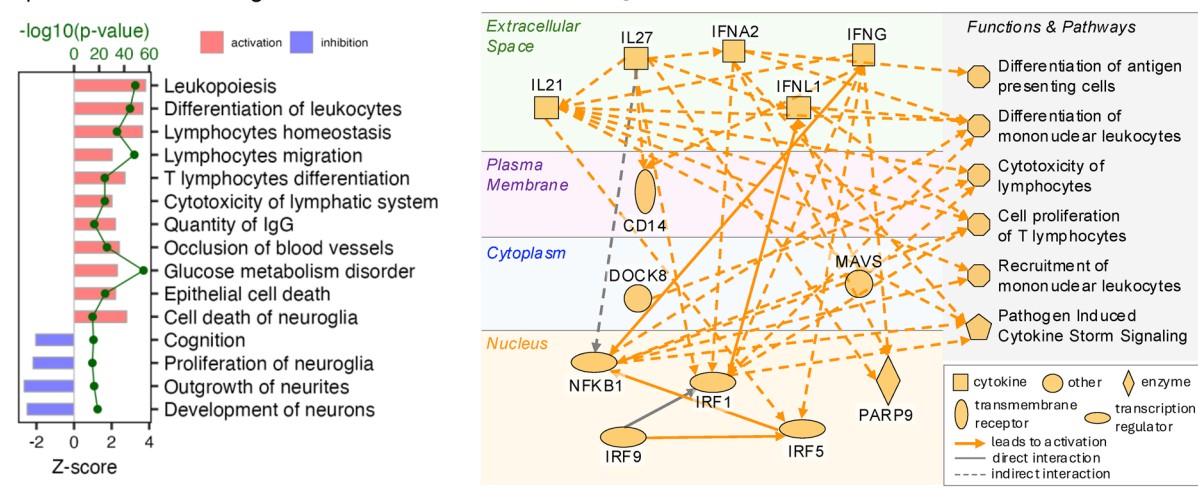

**d**   Graphical summary of IPA analysis of yearly change in proteins in longitudinal MS cohort

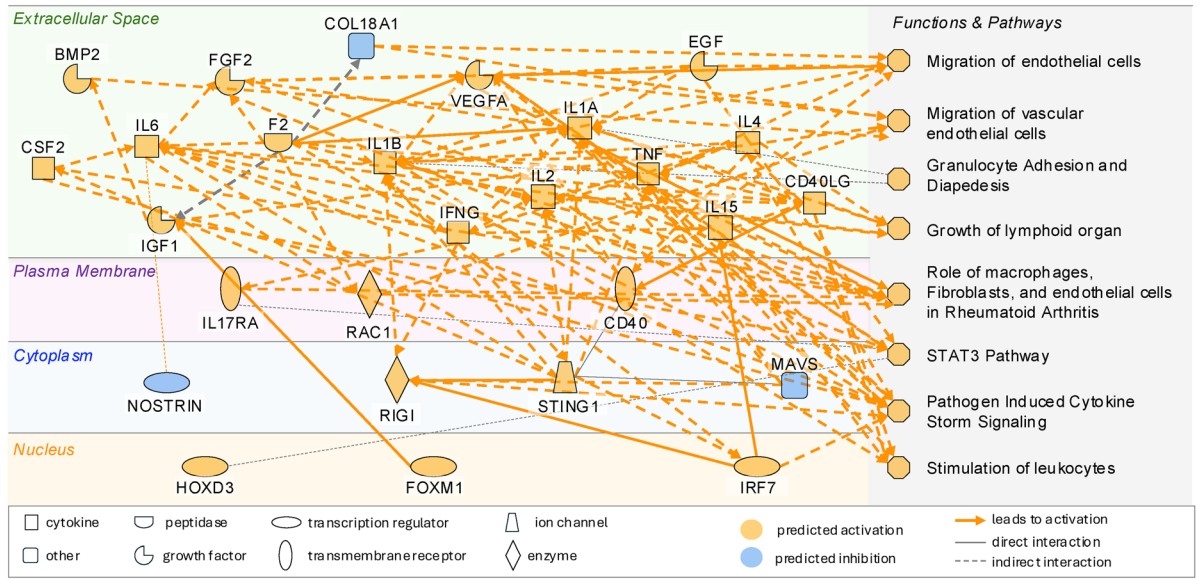

lymphatic endothelial cells (LEC; $p = 0.0114$), and fibroblast-derived proteins ($p = 0.00446$; Fig. 3d). Global decrease of these proteins in MS vs HC CSF reflects either loss of these cells or their physiological functions. Not observing such cellular bias for VEC proteins, we examined them in detail: plasma membrane proteins (JAM2, DLL1, JAG1, KDR) mediating physiological functions such as NOTCH signaling correlated negatively with CEL# (enrichment $p = 0.00221$-$0.0354$),

while positively correlating proteins were inflammatory biomarkers ITGA1/ITGB1, MMP7, IL6, and C9.

Thus, CSF biomarkers link CEL formation to proteins associated with blood-brain barrier (BBB) injury, intrathecal activation of anti-viral immunity, and oligodendroglial and neuronal loss. Dysregulation of LEC proteins may reflect tissue remodeling associated with phenotypic change[41].

**Fig. 2 | Proteomic data and summary models that differentiate untreated multiple sclerosis (MS) cohort from healthy controls (HC) and that evolve with MS natural history. a** Volcano plot of 270 somamers with cell-specificity to oligodendrocytes, monocytes, macrophages, and plasma cells that passed false discovery rate (FDR)-adjusted $p < 0.05$ in differentiating MS from HC. Differences were tested with two-sided Wilcoxon rank-sum test. Insets show protein PIGR (decreased in untreated MS, $n = 634$ vs HC, $n = 69$) and TNFRSF13B (increased in MS), evaluated by two-sided $t$-test (unadjusted $p$ values shown). Green lines mark HC means. Boxplots display medians (black line), quartiles, whiskers (1.5× interquartile range [IQR]), and outliers. One-sided chi-squared tests revealed over-representation of oligodendrocyte markers among proteins decreased in MS (blue circles) and immune-specific markers among proteins increased in MS (red circles). **b** Ingenuity Pathway Analysis (IPA) functions identified from proteins significantly differentiating MS from HC. Red bars denote activated processes, blue bars denote inhibited processes (based on z-scores). Green dots indicate significance level (unadjusted $-\log_{10}$ $p$-values). **c** IPA-derived summary model illustrating biological processes and pathways associated with proteins distinguishing MS from HC. **d** IPA summary model of proteins showing significant non-zero yearly change in longitudinal untreated MS cohort, reflecting disease evolution.

## The extent of BBB injury determines CEL destructiveness. Male pwMS form more destructive lesions with higher leukocyte extravasation and ECM degradation

To identify likely pathogenic mechanisms of lesional inflammation, we used CSF neurofilament light chain (NEFL) to measure CEL-associated axonal injury[42–44]. Because we abandoned research use of MRI contrasts after finding that gadolinium accumulates in brain[45], many CSF samples lacked associated post-contrast MRIs. To analyze all samples, we constructed a CSF biomarker-based model of CEL#, deliberately excluding NEFL (and related NEFH) from the modeling. CSF-predicted CEL# model reproduced MRI CEL# attributes such as age-related decline and NEFL correlations (Fig. 4g, h, Supplementary Fig. 2).

We restricted the analysis to participants with CSF-predicted CEL# above upper HC 95% prediction interval. We then quantified CEL-associated axonal injury using residuals from a linear regression of CSF NEFL concentration (y) on CEL count (x), such that higher residuals indicate more destructive lesions (Fig. 4i). Only 24 proteins correlated positively with both CEL# and CEL-NEFL residuals, 83% of which were cell-specific (Fig. 4j). Reassuringly, NEFH and (alternatively measured) NEFL were among them. ExNeuron-enriched intracellular proteins, released upon membrane disruption, were overrepresented among proteins linked to CEL destructiveness ($p = 0.0415$), as were the monocyte proteins ($p = 0.00328$; Fig. 4j). 46% of these proteins and 5 pathways (e.g., Degradation of ECM, Hepatic fibrosis, and Leukocyte extravasation) were increased in male pwMS.

Sixty proteins correlated negatively with CEL#, but positively with CEL-NEFL residuals, indicating reduced release in more destructive CELs (Fig. 3k). These included VEC- and fibroblast-derived proteins, with the latter significantly biased ($p = 0.0172$). Sex influenced 79% with greater CEL destructiveness in males.

Notably absent were biomarkers of adaptive immunity; instead, lesion destructiveness scaled with BBB injury. Males exhibited more destructive CELs, mirroring greater leukocyte extravasation and ECM-degradation. Whether monocytes drive this injury or merely accumulate to phagocytose debris can be answered only through interventional trials.

## Cumulative white matter injury measured as T2LL reflects loss of neuronal, glial, and epithelial cell markers and develops similarly between male and female pwMS

Of 818 proteins correlating with T2LL (Supplementary Fig. 3, Supplementary Note 2), only 14% were influenced by sex (Fig. 3) and just 12.7% also correlated with CELs. Proteins positively associated with both CEL# and T2LL were released by B-, T-, NK cells and monocyte/macrophages (Supplementary Fig. 3a). Both outcomes shared downmodulation of oligodendrocyte- ($p = 6.02e−8$), ExNeurons- ($p = 5.69e−13$), astrocyte- ($6.67e−04$) and LEC-specific proteins ($p = 8.01e−4$). PwMS with high T2LL also showed reduced InhNeurons- ($p = 6.02e−8$), and VEC-enriched proteins ($p = 6.64e−4$; Fig. 3). Downmodulated oligodendrocyte proteins localized to the Node of Ranvier ($p = 0.049$), paranodes- ($p = 0.0168$), and axo-glial junctions ($p = 0.0024$).

25% (17/69) of ExNeuron proteins reduced in patients with high T2LL were also globally decreased in MS CSF. These included glutamatergic synaptic proteins ($p = 0.007$), implying synaptic loss with T2LL accumulation. This was replicated for AMPA- ($p = 0.006$) and GABA-ergic synapses ($p = 0.006$) of InhNeurons. Unexpectedly, InhNeurons-downmodulated proteins were linked to depression phenotype ($p = 0.0033$) and BMI ($p = 0.0016$). While lacking depression data, we confirmed that 65% of these InhNeuron proteins were downmodulated in obese pwMS.

Thus, CSF biomarkers link T2LL accumulation to chronic inflammatory CNS injury with demyelination and neuronal/synaptic loss. Loss of GABA-ergic signaling may underlie associations of obesity with depression[46] and depression with MS[47].

## Stromal cell responses to CNS injury accompany increasing cognitive disability

Neurological disability has two forms: 1. cognitive, associated with brain volume loss measured as declining brain parenchymal fraction (BPFr) and psychomotor slowing (e.g., on Symbol-digit modalities test, SDMT); and 2. physical disability, assessed by neurological examination translated to disability scales. Only the most sensitive/specific outcomes can detect the subtle effects a protein may exert on complex phenotypes like disability. We therefore developed and applied optimized outcomes to maximize accuracy (Fig. 5).

Brain damage (BD), a composite of BPFr and SDMT, serves as a cognitive disability outcome that correlates strongly with both components ($R^2 > 0.71$, $p < 2.2e−16$) while minimizing noise. Of 819 BD-correlating proteins, 243 (30%) also correlated with T2LL, with 98% showing congruent directionality (Fig. 6a). 11% of BD-correlated proteins overlapped with CEL# correlates, mostly with opposing directionality. Just 21 biomarkers, representing 17 proteins, positively correlated with both CEL# and BD. These were enriched in myeloid cells ($p = 0.0127$) and involved in cell surface interactions at the vascular wall ($p = 0.0067$, Supplementary Note 3).

It appears that most CEL-associated processes do not drive cognitive disability. No immune cell type showed bias towards likely pathogenic or protective roles in BD accumulation (Fig. 3). ExNeuron proteins were preferentially decreased in people with high BD ($p = 0.0128$), suggesting that CEL/T2LL-related neuronal loss contributes to cognitive decline. In contrast to CEL and T2LL, BD accumulation was associated with increased fibroblast proteins ($p = 0.000579$; Fig. 6b). Fibroblast-enriched proteins reversed their associations: negatively correlated with CELs, but positively with CEL destructiveness and all disability outcomes (Fig. 3).

CNS stromal cells include epithelial cells (VEC, LEC, ependyma, and choroid plexus epithelium), pericytes, smooth muscle cells, and fibroblasts (Supplementary Note 1). Fibroblasts, enriched in meninges, extend along penetrating vessels into perivascular spaces[48]. Though not part of the immune system, stromal cells critically regulate inflammation and mediate tissue remodeling, including TLFs formation in MS meninges[8]. They serve as precursors to follicular dendritic cells (FDC) in TLFs, which recruit B cells via CXCL13 and present captured immune complexes (IC), promoting differentiation into PB/PC[49]. Underrepresented in

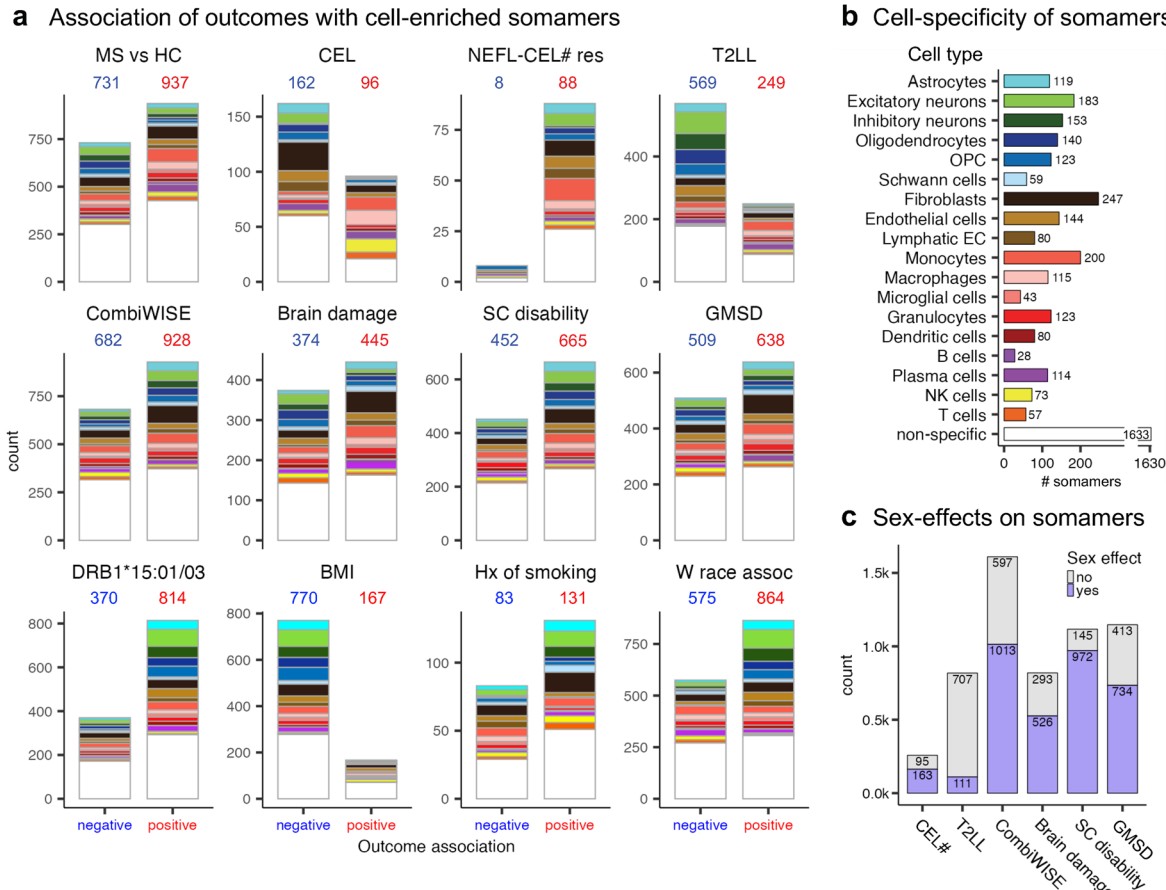

**a** Association of outcomes with cell-enriched somamers

**b** Cell-specificity of somamers

**c** Sex-effects on somamers

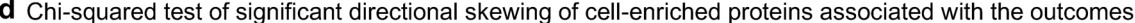

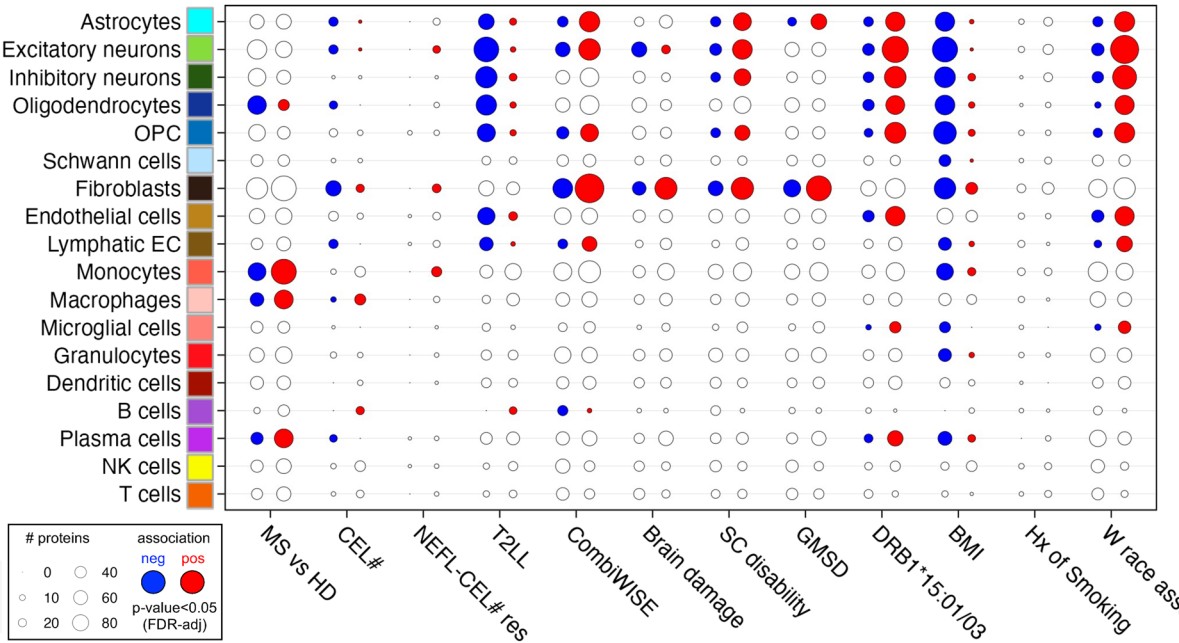

**d** Chi-squared test of significant directional skewing of cell-enriched proteins associated with the outcomes

snRNAseq data, CNS stromal cells remain understudied. IPA linked BD-correlating proteins to connective tissue cells ($p < 2.24\text{e}{-}15$), osteoclasts (myeloid cells participating in ECM remodeling; $p < 1.66\text{e}{-}13$), and vaso-occlusion ($p < 1.41\text{e}{-}20$), implicating stromal remodeling in BD accumulation.

64% of BD-correlating proteins differed by sex (Figs. 3 and 6c), with a strong pathogenic bias in males (Supplementary Note 3).

We conclude that while neuronal loss links CEL/T2LL with brain atrophy and cognitive disability, BD development is uniquely associated with stromal cell-mediated tissue remodeling. In males, increased ECM degradation and activation of complement and coagulation cascades are linked to faster brain atrophy[50]. Conversely, higher CSF levels of neurodevelopmental proteins negatively correlating with BD may reflect female-enriched resilience or repair mechanisms.

**Fig. 3 | Cell-enriched cerebrospinal fluid (CSF) proteins associated with multiple sclerosis (MS) outcomes and residual sex effects in untreated people with MS (pwMS). a** Associations of 12 clinical and demographic outcomes with healthy control (HC) age- and sex-adjusted somamer levels. Differences between MS and HC cohorts were tested using a two-sided Wilcoxon rank-sum test with false discovery rate (FDR)-adjusted *p*-values. For other outcomes, linear regression models with outcome, sex, and outcome:sex interaction terms were applied. Bar plots display the number of negatively (blue) and positively (red) associated somamers, with color indicating cell specificity (**b**). **b** Cell specificity of somamers was assigned using publicly available reference datasets (see Methods). **c** Numbers of somamers associated with six outcomes representing MS disease processes, with or without significant sex effects, were determined using regression models including sex and outcome:sex interaction. **d** One-sided chi-squared tests identified enrichment of cell-specific proteins associated with the 12 outcomes. Circle size represents the number of somamers; empty circles indicate non-significant associations; blue and red filled circles indicate significant negative and positive associations, respectively (FDR-adjusted *p*-values). CEL# – number of contrast-enhancing lesions; NEFL-CEL# res – neurofilament light chain residuals of CSF-predicted CEL#; T2LL – T2 lesion load; CombiWISE – Combinatorial Age-adjusted Disability Score; SC disability – spinal cord disability; GMSD – global MS disability; HLA-DRB1*15:01/03 – *human leukocyte antigen DRB1*15:01/03 haplotype; BMI – body mass index; Hx of smoking – history of smoking; W race assoc – White race association; OPC – oligodendroglial precursor cells; EC – endothelial cells; NK cells – natural killer cells.

## Tissue remodeling by collaborating myeloid and stromal cells accompanies T2LL-independent accumulation of cognitive disability

Neurodegenerative mechanisms are hypothesized to cause neuronal loss after MS lesions stop forming. To identify inflammation-independent neuronal injury, we used propensity score matching to compare pwMS with similar T2LL but divergent BD (Supplementary Fig. 4a), and another subgroup within the same cohort with similar BD, but differing T2LL (Supplementary Fig. 4b).

46 biomarkers (Supplementary Fig. 4c) and 3 pathways (Hepatic fibrosis, LXR/RXR activation, and HIF1α signaling) correlated positively with BD, were upregulated in pwMS with disproportionally high BD, and downmodulated in those with low BD. Among 23 cell-specific proteins, 39% originated from stromal cells and 17% from myeloid cells. IPA linked myeloid cells/phagocytosis ($p < 1.6e{-}13$), fibroblasts/connective tissue functions ($p < 3.44e{-}18$), but also viral infections ($p < 1.67e{-}23$) to disproportionally high BD (Supplementary Fig. 4d).

These findings implicate tissue remodeling by collaborating myeloid and stromal cells, linked to hypoxia, in the T2LL-independent accumulation of cognitive disability.

## Activation of myeloid cells associated with the formation of reactive oxygen/nitrogen species (ROS/NOS) and proteomic signatures of cellular death are shared processes linked to the development of cognitive and physical disabilities

We used the continuous Combinatorial Weight Adjusted Disability Scale (CombiWISE[51]), which ranges from 0–100 and correlates strongly with the ordinal Expanded Disability Status Scale ($R^2 = 0.91$, $p < 2.2e{-}16$; Fig. 5b) to measure physical disability. Because most pwMS accumulate cognitive and physical disabilities congruently, BD and CombiWISE are moderately correlated ($R^2 = 0.35$, $p < 2.2e{-}16$; Fig. 5c). We leveraged this to identify shared mechanisms of disability progression (Global MS disability, GMSD) and distinguish them from mechanism(s) underlying disproportionally higher physical disability reflecting spinal cord (SC) predominant injury (Fig. 5c,d).

Of 1610 CombiWISE-correlating proteins, 33.5% also correlated with BD; nearly all with the same directionality, except PB/PC-enriched DEFB1 (Fig. 7a, b, Supplementary Note 4). These shared mechanisms are captured in 142 GMSD-correlating pathways (Fig. 7c).

Only two cell types showed biased associations with GMSD, both towards pathogenicity (Fig. 3). While fibroblasts exhibited a pathogenic bias across all disability outcomes, astrocytes were biased specifically towards physical disability, not BD.

We conclude that immunogenic cell death of neurons and glia, and proinflammatory activation of myeloid cells, their formation of ROS/NOS, and their collaboration with stromal cells are shared processes linked to cognitive and physical disability accumulation.

## SC injury dissociated from BD is linked to continuous formation of immune complexes (IC) and dysregulated CSF1/IL34 signaling

To determine whether mechanisms of SC and BD injury differ, we identified biomarkers correlating with SC disability and used propensity score matching to isolate proteins differentiating pwMS with similar BD but divergent SC disability, and vice-versa.

Beyond the pathogenic skewing of astrocyte proteins with physical/SC disability outcomes (Fig. 3), 63 proteins/pathways positively correlated with SC disability, were increased in pwMS with higher CombiWISE when matched for BD and were either lower, or weakly linked to BD (Fig. 7d). Surprisingly, 92% overlapped with proteins/pathways associated with BD accumulation disproportionate to T2LL (Supplementary Fig. 4), suggesting these reflect shared non-lesional CNS damage mechanisms rather than SC-specific pathology (see Supplementary Note 1).

After excluding CEL-correlating biomarkers to avoid spurious negative correlations, only 5 proteins inversely correlated with SC disability: VEGFA, IL34, FCGR2A (CD32; Fc receptor on phagocytic cells), monocyte-enriched immunomodulatory cytokine NAMPT, and CNS-specific ECM protein FLTR1 (Fig. 7d). Because we found FCGR2A transcriptionally upregulated in MS myeloid cells (Supplementary data 3), low CSF levels of FCGR2A likely reflect epitope masking by ICs and phagocytic clearance. Its negative correlation with age and longitudinal decrease in untreated pwMS further links chronic ICs formation to MS progression and physical disability.

IL34 was also globally decreased in MS CSF despite being transcriptionally upregulated in MS CNS cells that normally secrete it (Supplementary data 3), implying increased consumption. IL34 and CSF1 signal through CSF1R, but with different outcomes: IL34 maintains the microglial homeostatic phenotype[52], whereas CSF1 supports proinflammatory activation. IL34 also binds SDC1 (CD138; syndecan-1, enriched on PB/PC), and the CNS-specific receptor PTPRZ1, expressed in OPCs and involved in remyelination[53]. IL34 consumption by PB/PC, expanded in MS CNS, may thus divert it away from CSF1R and PTPRZ1, disrupting both microglial homeostasis and remyelination. Supporting this hypothesis, IPA predicted upstream inhibition of myelination-related MOG and MYCN in SC-injury predominant pwMS.

Thus, while T2LL-dysproportional brain atrophy and SC injury share proteomic signatures, pwMS with disproportional SC injury have additional CSF proteomic changes suggestive of sustained ICs formation and disrupted IL34/CSF1 signaling axis.

## Fibrosis-related processes and Schwann cell proteins released during SC injury correlate with rates of physical disability progression in the longitudinal MS cohort

While untreated longitudinal samples revealed processes evolving with MS natural history (Fig. 2d), we asked whether leveraging all longitudinal CSF samples and clinical/imaging data, including from treated pwMS, could identify biomarkers that predict or reflect rates of MS progression. As the most sensitive progression outcome (Supplementary Fig. 5), only the CombiWISE slope yielded significant biomarkers after multiple comparisons correction.

In 213 pwMS with initial untreated CSF, the median follow-up included 7 clinical assessments over 6.2 years (range 1.5–20.1), during which patients progressed by 1.45 CombiWISE units/year

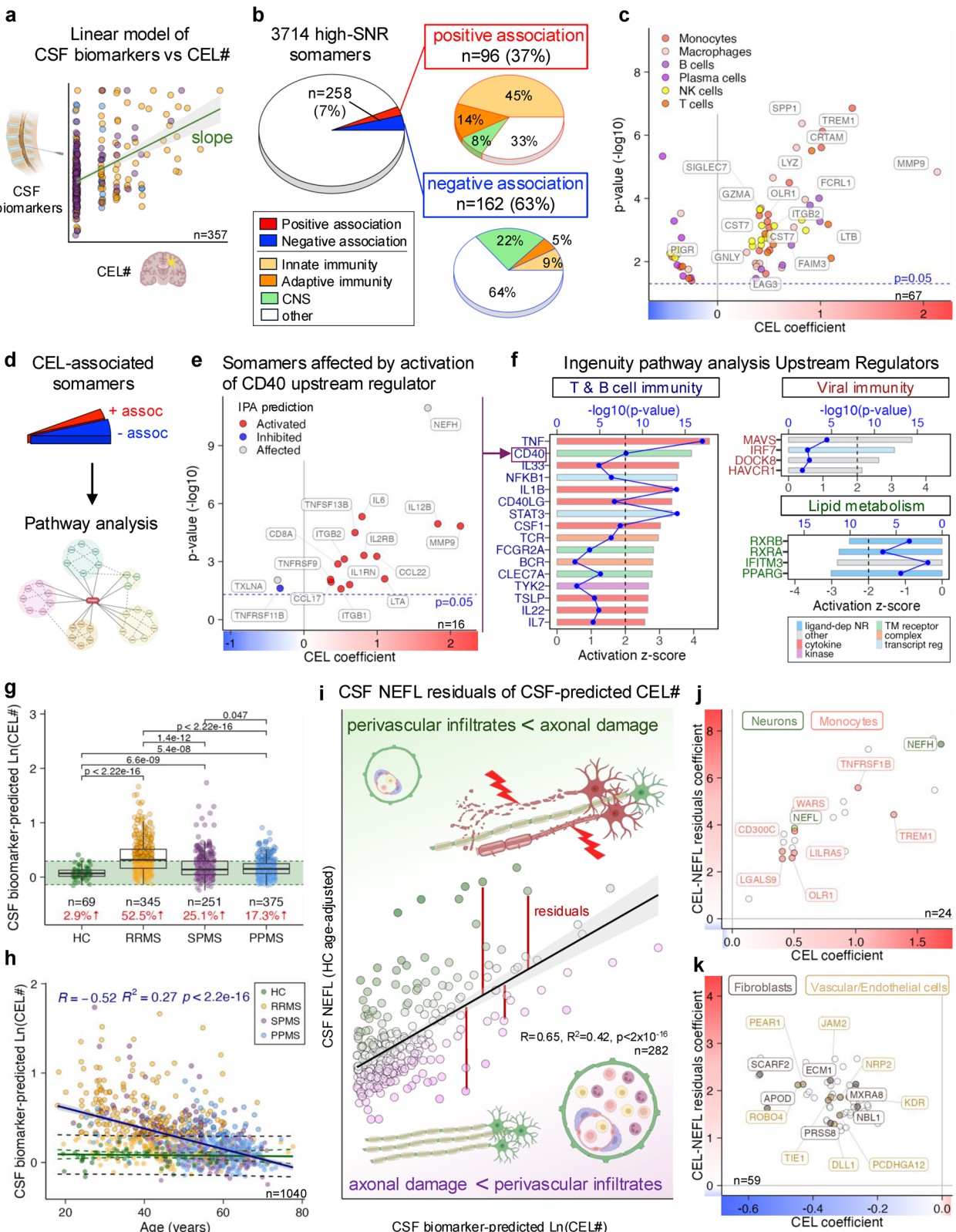

(p = 1.0e−39; Fig. 8a). Fastest progressors (3rd tertile; > 2.06 units/year; Fig. 8b) had significantly elevated levels of 6 proteins and 10 pathways compared to slowest progressors (1st tertile; <0.93 units/year; Fig. 8c). All 16 biomarkers were increased in MS CSF, suggesting pathogenic roles. They reflect activation of immune (especially myeloid) and stromal cells; 38% were influenced by sex, all increased in males.

We conclude that baseline biomarkers reflecting intrathecal inflammation and immune-stromal interaction can predict future disability progression.

We next asked whether yearly biomarker changes reflect yearly progression rates (Fig. 8e). Among 150 pwMS with at least two CSF samples (first untreated) collected ≥1.5 years apart (median 4.2, range 1.5–13.5) the group progressed by 1.67 CombiWISE units/year (p = 5.4e

**Fig. 4 | Cerebrospinal fluid (CSF) proteins, pathways, and predicted upstream regulators associated with contrast-enhancing lesions (CEL) and their destructiveness to central nervous system (CNS) tissue. a** Associations of CSF biomarkers with the number of CEL (CEL#) were assessed by linear regression, with CSF biomarkers as dependent variables and CEL#, sex, and CEL#:sex interaction as independent variables. Coefficients of CEL# characterized biomarker associations. Regression lines and 95% confidence intervals (CI) are shown (green with gray shading). **b** Seven percent of high-signaling somamers were significantly associated with CEL#, with one-third positive and two-thirds negative. Positive associations were dominated by immune-derived proteins, whereas negative associations were enriched for central nervous system–derived proteins. **c** Volcano plot of 67 proteins significantly associated with CEL# (unadjusted $p < 0.05$). Gray line shows coefficient = 0. Two CST7 labels reflect distinct epitopes of the same protein. **d** Somamers associated with CEL# were analyzed by Ingenuity Pathway Analysis (IPA), identifying pathways, upstream regulators, and networks. **e** Volcano plot of CEL#-associated somamers affected by CD40 upstream regulators. **f** Selected upstream regulators identified by IPA, shown as bar plots of bias-adjusted z scores with unadjusted −log10 $p$-values. **g** CSF biomarker-predicted CEL# from an elastic net model showed significant increases in people with multiple sclerosis (pwMS) compared with healthy controls (HC), and in relapsing-remitting MS (RRMS)

compared with primary progressive MS (PPMS) and secondary progressive MS (SPMS). The green rectangle shows HC mean ±1.5 standard deviation (SD); red % values indicate proportions above HC range. Significance was tested by two-sided one-way ANOVA with post-hoc t-tests. Boxplots display medians, quartiles, and whiskers (1.5× in interquartile range [IQR]). **h** Regression of biomarker-predicted CEL# and age showed CEL# decline with increasing age. Blue line with band shows regression with 95% CI; HC reference lines shown in green/black represent regression line with 95% CI/95% prediction interval. **i** Regression of neurofilament light chain (NEFL) levels and biomarker-predicted CEL# in 282 MS samples above the HC prediction interval was used to calculate NEFL−CEL residuals (dark red vertical lines), indicating axonal damage adjusted for CEL#. The regression line and 95% confidence interval are shown as a black line with a gray band. The regression model is characterized by the Pearson correlation coefficient (R), coefficient of variance ($R^2$), and unadjusted $p$-value. **j, k** Scatter plots showing somamers associated with CEL# and NEFL−CEL residuals, highlighting biomarkers from neurons (green), monocytes (salmon), fibroblasts (gray), and endothelial cells (beige). The solid gray lines mark coefficients of 0. All $p$-values of regression coefficients were tested in a two-sided test. Created in BioRender. Kosa, P. (2025) https://BioRender. com/g03z1b4.

−27; Fig. 8f, g). Yearly change in 57 unique proteins and 31 pathways differed between fastest and slowest progressors (Fig. 8h, i). Only the IL6 signaling pathway overlapped between predictors and reflectors of rapid progression. Again, sex effects were biased (77% increased) towards greater predicted pathogenicity in males.

Positive correlations (Fig. 8i) further support the likely pathogenic role of fibrotic processes in MS progression, while negative correlations support the beneficial role for remyelination-promoting RAR-, FXR/RXR pathways[54,55]. Intracellular Schwann cell proteins, including peripheral myelin protein MPZ (Supplementary Note 6) were highly represented. This likely reflects immunogenic Schwann cell death and demyelination at SC injury sites, as Schwann cells are typical cell population in the adult human spinal cord[56].

Thus, CSF biomarkers can both reflect and predict the rate of subsequent physical disability accumulation in MS.

Finally, the extensive intrathecal molecular effects of MS confounders, race, BMI, smoking, and DRB1*15:01/03 genotypes, are described in Supplementary notes 7–10 and Fig. 9.

## Discussion

Linking thousands of proteins from all inflamed CNS cell types to clinical and imaging data from hundreds of untreated pwMS, spanning the full MS phenotypic spectrum and incorporating covariates, makes this study uniquely complementary to MS pathology and transcriptomics studies. By integrating proteomic and transcriptomic data and interpreting them in the light of current knowledge, we offer here the most plausible mechanistic explanations, with additional data available in Supplementary Worksheets and an interactive website (https://bielekovalab0proteomics.shinyapps.io/shiny/). However, interpretation of the directionality remains uncertain, as transcriptomic signatures do not always reflect protein levels. Moreover, protein release can occur via secretion, cleavage of membrane proteins, and non-physiological release during immunogenic cell death. Conversely, consumption lowers levels through receptor binding, internalization, phagocytosis, or epitope masking (e.g., in ICs). Because upstream processes are varied and knowledge remains incomplete, all interpretations should be viewed as hypothesis-generating.

We consider these the key findings of the study:

First, proteins linked to viral infections and antiviral immunity were repeatedly prominent across analyses. Activation of all three interferon types, cytoplasmic viral sensors, and interferon-induced transcription factors persisted throughout MS natural history. Because signatures of anti-viral immunity correlate with favorable outcomes, we hypothesize that EBV may contribute not only to MS

onset[26] but also to its progression, although pathology evidence remains debated[57–61]. Our finding that increasing IC formation parallels MS evolution and disability progression could help account for the paradoxical decrease in anti-EBV antibodies[62] in MS CSF[63,64], even though at MS onset, anti-EBV antibodies overlap with MS oligoclonal bands[62]. Such depletion of antibodies due to IC binding has been described[65].

Second, MS confounders, particularly residual sex effects (beyond those subtracted using HC controls), strongly modulate candidate pathogenic and repair processes. Males exhibit greater proinflammatory tone, increased epithelial barrier injury, greater neuronal damage, and faster disability accumulation. In contrast, females appear to re-activate neurodevelopmental programs that normally decline with age in HC, likely contributing to tissue repair and neuronal maintenance.

Third, we identify a critical role for CNS stromal cells in MS evolution. While CEL# declines over time, proteomics indicates evidence of ongoing (potentially EBV-mediated[66–68]) injury to CNS epithelial barriers and associated increase in fibroblast- and ECM/fibrosis-related proteins in CSF. Fibroblast-enriched proteins switch from negative associations with CELs to positive associations with disability outcomes, suggesting a compensatory stromal response to limit tissue damage and maintain extracellular homeostasis. However, excessive fibrosis could impair oxygen[69] and nutrient delivery, while FDC differentiation and TLFs formation may retain EBV by providing survival niches for B cells and enabling periodic reactivations[70]. A unified hypothesis is presented in Supplementary Note 11 and abbreviated in Fig. 10.

The study has the following limitations: 1. Lack of an external validation cohort. To our knowledge, a comparably large and diverse longitudinal cohort of deeply phenotyped pwMS, including all three MS subtypes, most of whom donated CSF during the untreated stage, does not exist. This limits the ability to externally validate our findings. 2. Focus on untreated natural history rather than treatment effects. Because most CSF samples in this study were collected during the untreated stage of disease, the study is underpowered to assess the impact of therapies. We therefore focused on the natural history of untreated MS. However, we continue to collect CSF samples during the treatment period from pwMS who consent to donate, with the goal of eventually enabling rigorous analyses of treatment effects.

This resource paper does not answer all MS-related questions but provides a valuable dataset linking CSF proteomics with major aspects of MS biology, including the influence of key confounders, to support future mechanistic studies.

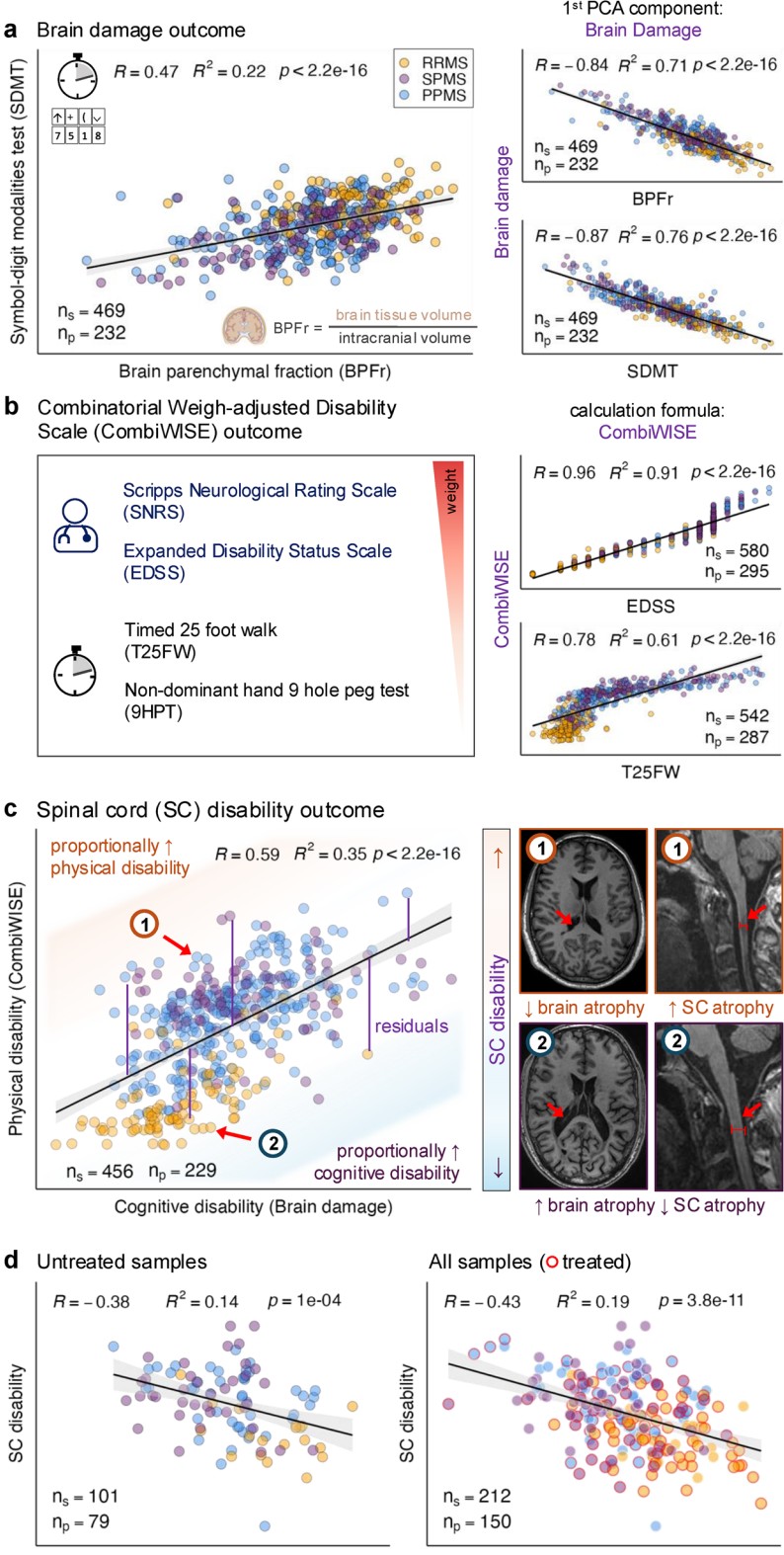

## Methods

### Study participants and CSF samples

This was a retrospective analysis of a cohort of 394 pwMS (124 with primary progressive MS [PP-MS], 179 with relapsing-remitting MS [RR-MS], and 91 with secondary progressive MS [SP-MS]) and 44 HCs that were prospectively enrolled between January 1999 and August 2024 into natural history protocol Comprehensive Multimodal Analysis of Neuroimmunological Diseases of the Central Nervous System (Clinicaltrials.gov identifier NCT00794352) approved by NIH Intramural Institutional Review Board (samples collected before 2009 were part of the NIB Repository Protocol [10-N-0210]). NIH Clinical Center is nationwide biomedical research hospital funded by USA federal government. It recruits participants interested to participate in research from the entire USA.

**Fig. 5 | Characterization of composite cognitive and physical disability outcomes in relation to traditional outcomes used in multiple sclerosis (MS) clinical trials. a** The brain damage outcome was generated as the first component of a principal component analysis (PCA) of brain parenchymal fraction (BPFr; x-axis, volumetric magnetic resonance imaging [MRI] ratio of brain tissue to intracranial volume) and Symbol-digit modalities test (SDMT; y-axis). Brain damage correlated strongly with both measures (plots on right). **b** The Combinatorial Weight-adjusted Disability Scale (CombiWISE), a linear combination of four disability scores (weights decreasing as shown by the red triangle), correlated strongly with traditional MS disability scales, the Expanded Disability Status Scale (EDSS) and Timed 25-foot walk (T25FW) (plots on right). **c** The spinal cord (SC) disability outcome was calculated as residuals of a linear regression model of CombiWISE (y-axis) versus

brain damage (x-axis). For similar brain damage, positive residuals (orange) indicated proportionally more physical disability, while negative residuals (blue) indicated less. MRI examples show patients with (1) high SC disability (low brain atrophy, high SC atrophy) and (2) low SC disability (marked brain atrophy, minimal SC atrophy). **d** Quantitative spinal cord atrophy, measured as C1–C2 SC area (x-axis), correlated strongly with SC disability (y-axis) in untreated (left) and all samples (right; treated samples in red). Scatter plot associations are shown as regression lines with 95% confidence intervals (black line, gray shading) and characterized by Pearson correlation coefficient (R), coefficient of determination ($R^2$), and unadjusted $p$-values. $n_s$ = number of samples; $n_p$ = number of patients. All regression coefficients were tested using two-sided p values. Created in BioRender. Kosa, P. (2025) https://BioRender.com/l8nkay6.

All participants signed a written informed consent. The protocol recruited patients with known or suspected diagnosis of MS. A thorough diagnostic workup included full neurological exam, MRI of the brain, functional tests (e.g., timed 25-foot walk [T25FW], nine-hole peg test [9HPT], SDMT), and laboratory tests of blood and CSF. Each patient was followed for a minimum of one year (with optional follow-up LP). Seventy-eight percent of MS patients were untreated at the time of first LP (Fig. 1b). The inclusion criteria for the HC cohort were age 18–75, lack of neurological diagnosis or systemic disease that would influence neurological functions, or brain MRI and vital signs in the normal range during the initial screening.

A total of 1040 CSF samples were collected, represented by 69 CSF samples from HC and 971 CSF samples from MS patients, 634 (65.3%) of those in the untreated stage. Samples on treatment included daclizumab (26.7%), interferons (14.2%), rituximab (13.6%), glatiramer acetate (11.9%), natalizumab (8.0%), dimethyl fumarate (7.1%), fingolimod (3.9%), ocrelizumab (3.3%), alemtuzumab (1.2%), teriflunomide (1.2%), and other (8.9%).

The demographic data of all participants are detailed in Supplementary Table 1 and Fig. 1.

## Sample processing

CSF was collected on ice and processed according to a written standard operating procedure by investigators blinded to diagnoses, clinical, and imaging outcomes. Aliquots were assigned alphanumeric identifiers and centrifuged for 10 min at 300 $g$ at 4 °C within 30 min of collection. The cell-free supernatants were aliquoted and stored in polypropylene tubes at −80 °C. CSF and serum samples were sent to the NIH Department of Laboratory Medicine (DLM) for standard clinical tests (e.g., CSF IgG, serum CRP).

## MRI imaging

MRI of the brain was generated on 1.5 T and 3 T scanners (General Electric & Siemens). T1 magnetization-prepared rapid gradient echo (MPRAGE) and T2-weighted 3D fluid attenuation inversion recovery (3D FLAIR) sequences were obtained. Assessment of contrast enhancement was performed using postcontrast (gadopentetate dimeglumine at 0.1 mmol/kg) T1-weighted and postcontrast FLAIR images. The number of CEL was recorded in the research database. The MRI protocol extended caudally to the C5 level to allow analysis of the upper cervical SC.

The volumetric analysis was performed using the LesionTOADS volume segmentation algorithm[71] performed on the QMENTA imaging platform (www.qmenta.com). The details of the analyses have been described[72]. Brain parenchymal fraction (BPFr) was calculated as a ratio between the total brain volume and the intracranial cavity volume. Upper cervical SC cross-sectional area (C1–C2) was calculated from brain MRI images using Spinal Cord Toolbox[73].

## Clinical outcomes

Neurological exams have been recorded directly (after 9/2017) or transcribed retrospectively from structured medical records

neurological examination form (before 9/2017) into the NeurEx™ App[74] that automatically calculates traditional MS disability outcomes: EDSS[75] and SNRS[76]. CombiWISE was calculated from EDSS, SNRS, T25FW and nondominant hand 9HPT, as described[51] (Fig. 5a).

We have generated a set of new outcomes measuring both cognitive and physical disability. We combined MRI volumetric measure BPFr and psychometric SDMT into a principal component analysis, and we isolated the first component that, after inversion (so that increasing value corresponds to increasing disability) constituted the Brain damage outcome (Fig. 5b).

Taking advantage of the positive correlation between Brain damage and CombiWISE, we calculated a GMSD as a comprehensive marker of MS-related CNS injury by summing up normalized values of CombiWISE and Brain damage outcomes. Next, we isolated residuals of the CombiWISE-Brain damage linear model as SC disability outcome that differentiates patients with predominant cognitive (associated with brain damage) and physical (associated with spinal cord damage) disability (Fig. 5c). SC disability correlates with a fully quantitative MRI biomarker of spinal cord atrophy (Fig. 5d).

## ELISA immunoassays

CSF levels of NEFL were quantified using spectrophotometric assays by UmanDiagnostics (catalog# 10-7002) by personnel blinded to diagnostic and clinical/MRI outcome data. Details of the NEFL assay and all ELISA assays used for orthogonal validation of relative concentrations of selected CSF proteins measured by DNA aptamers (i.e., SomaScan) have been published[43,77]. Somascan-measured CSF biomarkers that originate in the serum (such as CRP) were orthogonally validated against identical proteins measured by NIH DLM. Finally, some Somascan-measured CSF biomarkers that originate from specific immune cells (e.g., CLEC4 originating from plasmacytoid DCs) were orthogonally validated against absolute numbers of their source cells enumerated per 1 ml of CSF using prospectively acquired multicolor flow cytometry on fresh CSF cells isolated from identical samples as described[78,79].

## SomaScan assay

A total of 1042 unique CSF samples marked with alphanumeric codes were analyzed by the SomaScan assay (Novartis V3-5K assay version, SomaLogic Inc, Boulder, CO, USA). Samples representing different diagnostic groups were interspersed between individual 96-well plates, while longitudinal samples of a single patient were kept within the same plate. SomaScan measures relative fluorescent units (RFU) of 4789 DNA aptamers, called somamers, that bind to 4114 human proteins. The raw RFUs have been mathematically processed to normalize the hybridization signal within each plate and to calibrate the signal across different plates, using control samples embedded within each plate. For details about published studies of orthogonal validation of the SomaScan assay, as well as extensive internal orthogonal validation of CSF biomarkers generated in the current study, see Supplementary Note 12. and Supplementary Fig. 6.

**a** Overlap of proteins associated with T2LL and Brain damage

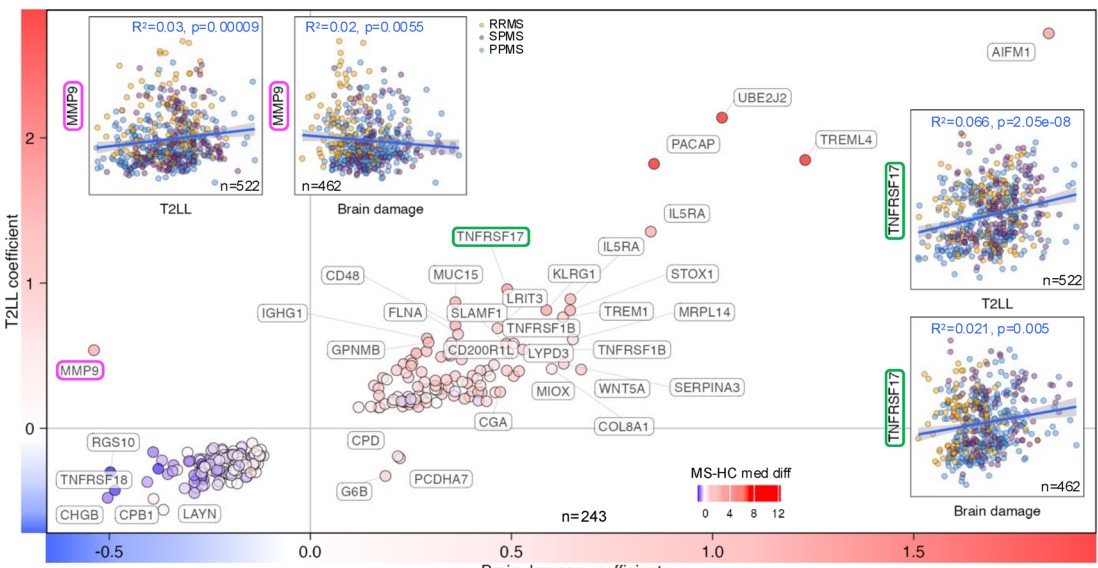

**b** Excitatory neurons and fibroblast-specific proteins significantly associated with Brain damage

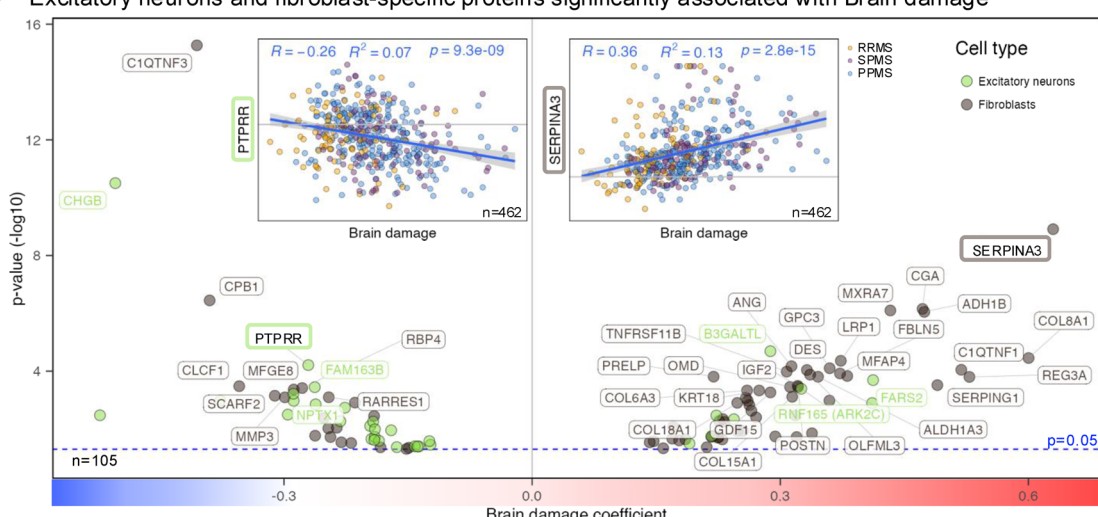

**c** Sex-effects on somamers associated with Brain damage outcome

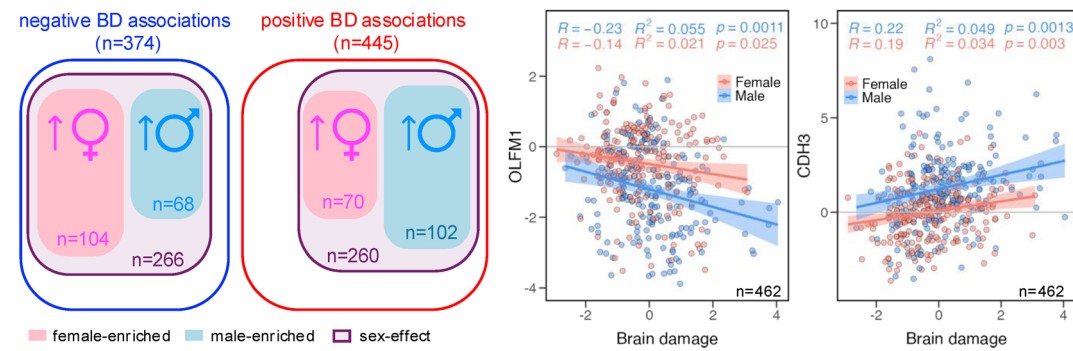

### Outlier analysis and signal-to-noise ratio calculation

Raw RFUs were first log-transformed (natural log). In the HC samples, we flagged all somamer values outside of the +/− 1.5 * interquartile range (IQR). Two HC samples with more than 15% of somamers flagged as outliers were removed from further analyses, resulting in a clean cohort of 1040 CSF samples.

Signal-to-noise ratio (SNR) was calculated as follows:

$$ \text{SNR} = \frac{\sigma^2_{\text{clin}}}{\sigma^2_{\text{clin}} + \sigma^2_{\text{tech}}} $$

where $\sigma_{\text{clin}}$ is the residual standard deviation of the linear model with just an intercept in the cohort of MS samples, and $\sigma_{\text{tech}}$ is the residual

**Fig. 6 | Cerebrospinal fluid (CSF) proteins associated with brain damage (BD) outcome reflecting cognitive disability. a** A total of 243 somamers significantly associated with BD (x-axis) and T2 lesion load (T2LL, y-axis) displayed congruent directionality in 98% of cases. Circle fill colors indicate median differences between multiple sclerosis (MS) and healthy control (HC) cohorts (red = elevated in MS, blue = decreased in MS). Examples include MMP9 (magenta) and TNFRSF17 (green), both correlated with T2LL and BD in the untreated MS cohort subdivided into relapsing-remitting MS (RRMS, yellow), secondary progressive MS (SPMS, violet), and primary progressive MS (PPMS, blue). Blue regression lines with gray 95% confidence intervals (CI) are shown, characterized by coefficient of determination (R²) and unadjusted p-values. Horizontal and vertical solid gray lines mark coefficients of 0. **b** Volcano plot of 105 somamers significantly associated with BD and enriched for excitatory neurons (green circles) or fibroblasts (dark gray circles). X-axes show BD coefficients, y-axes unadjusted −log10 p-values; the horizontal dashed blue line marks the cutoff (p ≤ 0.05). Inserts show examples: PTPRR

(negative BD association) and SERPINA3 (positive BD association) correlated with BD in untreated MS subgroups (RRMS, SPMS, PPMS). Regression lines and 95% CI are shown in blue/gray, characterized by R² and unadjusted p-values. The main plot includes gray lines for coefficient = 0 and p = 0.05 thresholds; inserts include HC-adjusted baseline levels at 0. **c** Many BD-associated somamers with negative (blue oval) or positive (red oval) correlations also showed residual sex effects (violet oval). Female-elevated proteins (pink oval) were more frequent among negative associations, while male-elevated proteins (blue oval) predominated among positive associations. Example scatter plots show OLFM1 (elevated in MS females) and CDH3 (elevated in MS males) correlated with BD. Blue (males) and pink (females) regression lines with 95% CI bands are shown, characterized by Pearson correlation coefficient (R), R², and unadjusted p-values. Solid gray lines indicate HC-adjusted somamer levels of 0. All p-values of regression coefficients were tested in two-sided test.

standard deviation of the linear mixed-effect model of somamer with patient-associated random effects in a longitudinal cohort of HC samples. The closer SNR is to 1, the more reliable the marker is. We used SNR cutoff of 0.8 and identified 3714 high-signaling somamers (Supplementary data 1).

### Adjustment for physiological aging and sexual dimorphism

To regress out the effect of natural aging and sexual dimorphism, we performed a two-step analysis. Because our HC dataset was relatively small, first, we borrowed information from the public domain, using two large studies of healthy individuals' plasma samples analyzed by previous versions of the SomaScan assays – Interval[80] and InChianti[81] study – and we identified somamers significantly associated with age and sex in HC. For identified somamers, we generated a multiple linear regression model with age and sex as predictors in our HC CSF dataset, and somamers that passed an unadjusted p-value cutoff of ≤ 0.05 and had the same directionality of the age/sex-association as those in the published studies were selected for adjustment, resulting in 103 somamers associated with age and 44 somamers associated with sex. Second, we hypothesized that some age/sex-associations might be matrix-dependent. Therefore, in our HC CSF samples, we generated multiple linear regression models of age and sex for all remaining somamers. Models with FDR-adjusted p-value below 0.05 were used to extract a list of an additional 206 somamers associated with age and 141 additional somamers associated with sex in HC samples. For identified somamers (309 for age and 185 for sex-adjustment), we subtracted values predicted in the HC cohort from measured values in all samples. For somamers not associated with age/sex, we subtracted the HC mean value.

The HC age/sex-adjusted somamers were then scaled to HC mean of 0 and standard deviation of 1.

Because the majority of the downstream analyses relied on normally distributed datasets, outliers presented a problem. However, unlike in the HC cohort, where it is reasonable to assume that outlier values are more likely of technical origin, in MS samples it is possible that outlier values are biologically meaningful. To retain values in the analyses but eliminate the problem of outliers, we floored the outliers below the 1st quartile − 3*IQR cutoff to the value of the cutoff, and for values above the 3rd quartile + 3*IQR cutoff to the value of the cutoff. This dataset was then used for all downstream analyses.

### Propensity matching

To identify the biology that differentiates brain atrophy vs the accumulation of T2LL, we generated two sub-cohorts of propensity score-matched samples. (1) Linear regression model between BD and T2LL identified BD residuals – patients with proportionally lower and higher measured BD than what would be predicted by their T2LL levels. Patients with BD residuals within IQR of the data distribution were

removed and the remaining samples with BD residuals lower than the first quartile (Q1) and greater than the third quartile (Q3) were matched for T2LL levels using propensity score matching (matchit function with full method; MatchIt R package[82]), resulting in Q1-Q3 paired samples with comparable levels of T2LL and significantly different levels of BD (Supplementary Fig. 4a) (2) We calculated T2LL residuals from a model using BD as independent variable and T2LL as depended variable and after removal of samples with T2LL residuals falling within the IQR of the data distribution we paired remaining samples between Q1 and Q3 group using propensity score matching, producing a cohort with comparable BD levels and significantly different T2LL levels (Supplementary Fig. 4b).

To identify biology that differentiates brain atrophy vs SC damage, we performed analogous analyses as described above, by substituting T2LL with CombiWISE, generating two sub-cohorts of propensity matched samples with 1) comparable levels of brain damage and significantly different levels of CombiWISE and 2) comparable levels of CombiWISE and significantly different levels of BD (see examples outside the x and y axes in Fig. 7d).

### Longitudinal analyses

We generated a longitudinal cohort of untreated MS samples by filtering patients with at least two CSF samples in the untreated stage separated by at least 1.5 years; in this cohort, we built linear models of protein vs age, and using a one-sample Wilcoxon test, we evaluated the null hypothesis of 0 median slope in the cohort. The raw p-values were FDR-adjusted for multiple comparisons.

Analyzing the relationship of CSF proteins to within-participant outcomes has much lower power than cross-sectional analyses due to practical considerations: 1. Limited cohorts, because longitudinal CSF sampling is available only for the most enthusiastic research participants. 2. Short follow-ups are associated with small (difficult to measure) changes, while long follow-ups are rare due to patient attrition. 3. Because most pwMS are treated during follow-up, heterogeneity of treatments and evolving comorbidities further limit power. Understanding these limitations, we performed 2 longitudinal analyses for each progression outcome. (1) We asked whether a biomarker correlating with the outcome cross-sectionally predicts subsequent longitudinal change in this outcome. We analyzed 219 pwMS who had their first LP in the untreated stage and a minimum of 1.5 years (median 6.2, range 1.5–20.1) of subsequent longitudinal follow-up that included a median of 8 clinic visits (range 2–36). For each patient, we calculated the yearly change for each outcome using a linear model of outcome versus age. Histograms of resulting progression slopes followed Gaussian distributions after removal of outliers using +/− 3*IQR cut-off. We then divided pwMS into progression slope tertiles and asked whether pwMS in the first versus the last outcome slope tertiles had statistically different biomarker concentrations at initial LP

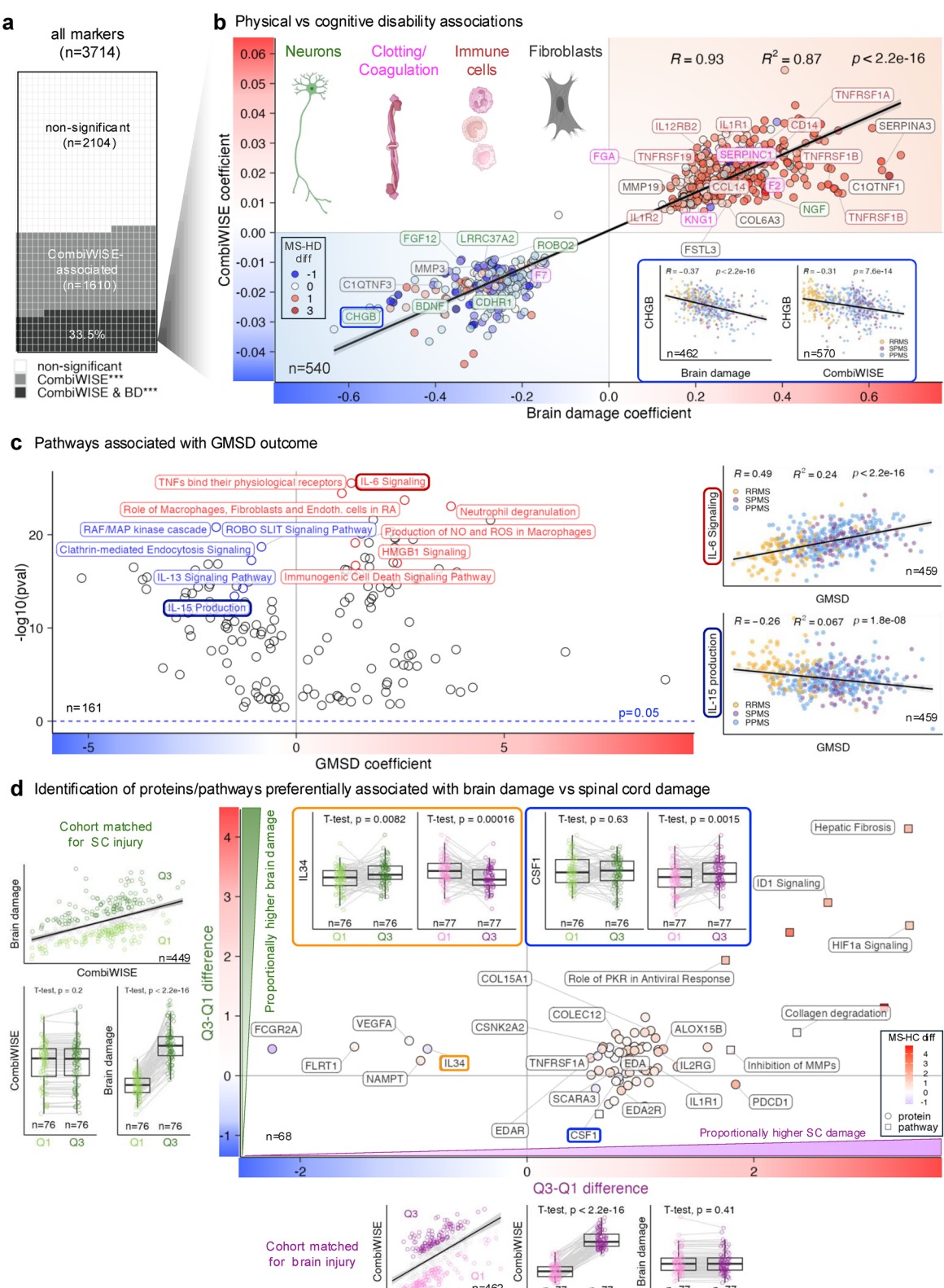

**a** all markers (n=3714)

**b** Physical vs cognitive disability associations

**c** Pathways associated with GMSD outcome

**d** Identification of proteins/pathways preferentially associated with brain damage vs spinal cord damage

(using two-sided Wilcoxon rank-sum test), with the biomarker-outcome directionality congruent with the cross-sectional analyses (Fig. 8a, b, Supplementary Fig. 5a, b). 2) The above longitudinal analysis benefits from a large number of patients and utilizes all available longitudinal outcome data. However, by not considering the interim biomarker change, the predictive analysis may disregard treatment-responsive proteins. We therefore generated

outcome progression slopes only from clinical visits occurring between the first and last LP in 154 pwMS with longitudinal LPs as described above. We then compared differences in yearly biomarker slopes in patients with slow (e.g., 1st tertile) versus fast (e.g., 3rd tertile) outcome accumulation, using two-sided Wilcoxon rank-sum test (Fig. 8e, f, Supplementary Fig. 5a,c). Raw *p*-values were FDR-adjusted for multiple comparisons.

**Fig. 7 | Proteins and biological processes associated with the development of physical disability. a** Diagram showing proportions of high-signaling somamers significantly associated with the Combinatorial Weight-adjusted Disability Scale (CombiWISE) alone (gray tiles) or with both CombiWISE and brain damage (BD) simultaneously (black tiles). **b** A set of 540 somamers significantly associated with CombiWISE (y-axis) and BD (x-axis) showed congruent directionality and strong correlation of outcome coefficients. Regression line with 95% confidence interval (CI) is shown as black line with gray band, characterized by Pearson correlation coefficient (R), coefficient of determination (R²), and unadjusted *p*-value. Examples include proteins linked to neurons (green), coagulation (magenta), immune cells (dark red), and fibroblasts (dark gray). Circle fill indicates median difference between multiple sclerosis (MS) and healthy control (HC) cohorts (red = elevated in MS; blue = decreased in MS). Inserts (blue oval) show negative correlations of CHGB with BD (left) and CombiWISE (right) in untreated MS cohorts split into relapsing-remitting MS (RRMS, yellow), secondary progressive MS (SPMS, violet), and primary progressive MS (PPMS, blue). Regression lines with 95% CI shown in black/gray. **c** Volcano plot of 161 pathways/clusters significantly associated with global MS disability (GMSD). X-axis shows GMSD coefficient; y-axis, unadjusted −log10 *p*-value. Negative associations (blue) and positive associations (red) are highlighted. Inserts show examples: IL-6 signaling (red) and IL-15 production (blue) pathways correlated with GMSD across MS subgroups (RRMS, SPMS, PPMS). Regression lines with 95% CI shown in black/gray. **d** Residuals from regression of BD (dependent) on CombiWISE (independent) identified two matched patient groups (Q1, light green; Q3, dark green) with similar physical but differing cognitive disability. Somamer differences between Q3 and Q1 are plotted on the y-axis. Similarly, residuals from the regression of CombiWISE on BD identified Q1 (pink) and Q3 (purple) groups with comparable BD but differing CombiWISE, with differences plotted on the x-axis. Higher x-axis values indicate proportionally greater spinal cord (SC) damage; higher y-axis values, greater cognitive disability. Proteins are circles; pathways are squares. Inserts highlight IL34 (orange frame) and CSF1 (blue frame) levels in matched groups. Paired two-sided *t*-tests assessed differences. Boxplots show medians, quartiles, and whiskers (1.5 × interquartile range [IQR]). All p values of regression coefficients were tested in a two-sided test. Created in BioRender. Kosa, P. (2025) https://BioRender.com/z9z0735.

## Knowledge base databases

To gain biological insight about molecular processes associated with different MS outcomes, we extracted somamers significantly associated with outcomes and uploaded them with the outcome coefficient values into Ingenuity Pathway Analysis (IPA®) and performed the Core Expression Analysis using Expr Other as a measurement type. For two-group outcomes (e.g., HC vs MS, propensity-matched groups) we used the median difference as the measurement type. The identified Canonical pathways, Upstream regulators, Causal networks, and Diseases and Biological functions were exported for each outcome, filtering those pathways/regulators/networks/functions with an absolute value of activation z-score ≥ 2. (Supplementary data 5–8).

Next, we took advantage of the IPA knowledge base that grouped and analyzed proteins into pathways and processes to generate patient-specific IPA pathway activation scores, hypothesizing that summing up the effects of individual proteins grouped by a biological process will amplify the signal of such a composite biomarker. For this purpose, we extracted Canonical pathways identified for GMSD outcome from IPA and filtered those with *p*-value < $10^{-5}$. IPA assigns a list of proteins from the query dataset to each Canonical pathway with their expected directionality. However, IPA predicts directionality of aggregated data (e.g., pathways) from transcriptomics datasets. Due to post-transcriptional regulations, diverse modes of protein release (e.g., secretion, transmembrane protein cleavage or cytoplasmic/nuclear/mitochondrial content released during immunogenic cell death) compounded by differences in protein consumption (e.g., receptor-ligand engagement, binding to ECM, phagocytosis), transcriptional directionality may provide an incorrect interpretation of processes CSF proteins reflect. Therefore, we used the directionality of CSF proteins with disability outcome to infer the directionality of IPA pathways. Specifically, we generated directionality information for all proteins based on their Spearman correlation coefficient with the GMSD outcome. Individual IPA pathway activation scores were calculated as the sum of scaled HC age-/sex-adjusted values multiplied by directionality (e.g., 1 or −1, based on their correlation with GMSD) for all somamers identified by IPA, resulting in 107 progression IPA pathway activation scores.

Analogously, we generated 61 MS vs HC IPA pathway activation scores, based on the IPA results from the analysis of somamer differentiating between MS and HC groups. In this instance, the directionality for each somamer was based on the sign of the difference between MS and HC group somamer levels.

IPA knowledge base depends on the curated source of data and, therefore, it can't be used for unbiased analyses. We hypothesized

that proteins that were not identified by IPA as being part of any known pathways/processes might still be important in CNS-specific biology and that proteins that work together will most likely correlate in a cross-sectional cohort of samples. Therefore, we generated a correlation matrix of proteins that 1) were significantly associated with GMSD and 2) were not used in any of the known IPA pathway activation scores. We isolated the pair of the strongest correlated somamers (the absolute Spearman Rho ≥ 0.7) that served as a seed for a new cluster. Then we kept adding more somamers to the cluster, as long as the new somamer retained an absolute Spearman correlation coefficient ≥ 0.5 with all existing members of the cluster. This process resulted in the generation of 4 progression clusters. The patient-specific progression cluster scores were calculated as described above for the progression IPA pathway activation scores. Similarly, we generated 10 new MS vs HC clusters, this time clustering somamers (as described above) that significantly differentiate MS from HC and were not used in any of the MS vs HC IPA pathway activation scores.

The composition of each of the 182 pathway/cluster scores is detailed in Supplementary data 9-10.

In addition to IPA, we used the following public databases to help us interpret the results of our analyses: Human protein atlas (https://www.proteinatlas.org), GTEx portal (https://gtexportal.org), STRING database (https://string-db.org), and Allen Brain Map (https://portal.brain-map.org).

We used an RNA single-cell type dataset downloaded from the Human Protein Atlas (that summarizes transcript expression levels [as normalized transcripts per million – nTPM] per gene in 81 cell types in 31 datasets). We matched the transcription information with our dataset for cell types relevant for the CNS and immune system (Astrocytes, B cells, Dendritic cells, Endothelial cells, Excitatory neurons, Fibroblasts, Granulocytes, Inhibitory neurons, Lymphatic endothelial cells, Macrophages, Microglial cells, Monocytes, NK cells, OPC, Oligodendrocytes, Plasma cells, Schwann cells, and T cells). For each protein, we identified cell type with the maximum nTPM and calculated the median nTPM across all cell types. To assign cell-specificity to a biomarker, the maximum nTPM/median nTPM must be > 5, and the maximum nTPM must be > 5.

## Elastic net (EN) model

We generated the EN model predicting the number of CEL (CSF predicted CEL#) using HC age-/sex-adjusted somamers in a training cohort of 394 MS CSF samples using the glmnet R package[83]. The best alpha (a coefficient that controls the balance between Lasso [alpha = 1] and Ridge [alpha = 0] regularization) was selected from a vector of 20

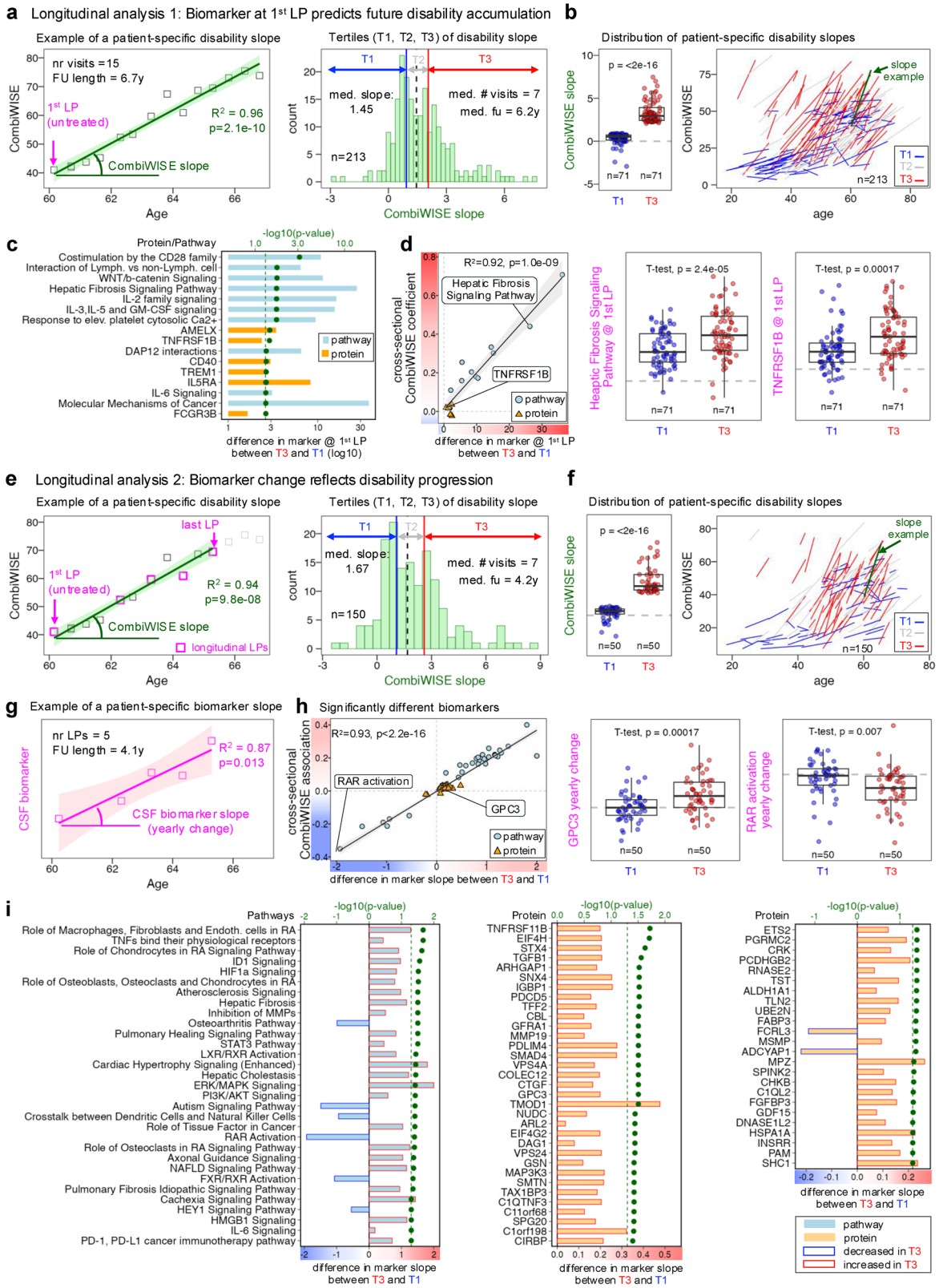

**a** Longitudinal analysis 1: Biomarker at 1st LP predicts future disability accumulation

**e** Longitudinal analysis 2: Biomarker change reflects disability progression

values between 0 and 1 based on the lowest mean squared error (MSE) calculated from 10-fold cross-validation. The performance of the model based on the best alpha was then tested in an independent cohort of 114 CSF samples, assessing the Spearman correlation coefficient (Rho), coefficient of determination ($R^2$), and Concordance Correlation Coefficient (CCC) between measured and predicted number of CEL (Supplementary Fig. 2). The details of the model and

results of the STRING analysis of the CSF predicted CEL# predictors are in Supplementary data 11-12.

## Reanalysis of single-nucleus brain tissue RNA sequencing datasets

We used published snRNA-seq datasets from MS and control brain tissue (GSE180759[15], GSE118257[16], GSE227781[17] and PRJNA544731[18,19]).

**Fig. 8 | Longitudinal analyses identify proteins that predict or reflect rates of disability accumulation in multiple sclerosis (MS). a** Longitudinal analysis of baseline biomarkers predicting disability accumulation. Left: example patient followed for 6.7 years (15 visits) showing positive correlation between disability measured by Combinatorial Weight-adjusted Disability Scale (CombiWISE, y-axis) and age (x-axis). First untreated lumbar puncture (LP) indicated by magenta arrow. The regression line with 95% confidence interval (CI) is shown (green/gray), characterized by the coefficient of determination ($R^2$) and unadjusted *p*-value. Right: histogram of CombiWISE slopes in 213 patients, split into tertiles (T1, blue = slow accumulation; T3, red = fast accumulation). **b** Left: significant differences in CombiWISE slopes between T1 and T3 patients (Wilcoxon rank-sum test unadjusted *p*-value). Right: individual patient slopes showing separation of slow vs fast accumulators; patient from panel a highlighted in green. **c** Bar plot of proteins (orange) and pathways (blue) at first LP predicting disability accumulation. Green dots show false discovery rate (FDR)-corrected −log10 *p*-values from Wilcoxon tests comparing T3 vs T1. **d** Correlation between baseline biomarker differences (T3 vs T1) and cross-sectional CombiWISE associations, shown for proteins (orange triangles) and pathways (blue circles). Regression line with 95% CI (black/gray) characterized by $R^2$ and unadjusted *p*. Examples (Hepatic fibrosis signaling pathway, TNFRSF1B) shown in boxplots comparing T1 and T3 with unadjusted two-sample t-test *p*-values. **e**, **f** Longitudinal biomarker changes reflecting progression. Left: example patient (same as **a**), slopes calculated only between first untreated and last LPs (magenta squares). Right: histogram of CombiWISE slopes in 150 patients, with T1 vs T3 groups. Significant slope differences are shown (Wilcoxon test). Individual patient slopes highlighted (patient from **e** in green). **g** Regression of CSF biomarker (y-axis) vs age (x-axis) used to calculate biomarker slopes (yearly change). Magenta regression line with 95% CI shown. **h** Correlation between biomarker slope differences (T3 vs T1) and cross-sectional CombiWISE associations for proteins and pathways, with examples (GPC3, RAR activation pathway) showing unadjusted two-sample t-test *p*-values. **i** Bar plots of pathways (blue) and proteins (orange) with yearly changes differing between T3 and T1. Biomarkers decreased in fast progressors outlined in blue; increased in red. Green dots represent FDR-corrected -log10(*p*-value) of the two-sample Wilcoxon test assessing the null hypothesis of no difference in biomarker yearly change between patients in T3 and T1 groups. Boxplots in panels b, d, f, and h show medians, quartiles, and whiskers (1.5 × interquartile range [IQR]). All statistical tests were two-sided.

---

27 pwMS and 23 controls were included (Age: 50.6 ± 10.7 vs. 57 ± 15.1 years old, female proportion 12/27 (44%) and 8/23 (35%), respectively). Mean disease duration is 19.3 ± 8.8 years and included 2 PPMS and 25 SPMS patients.

Each raw data set was counted, and QC was checked by Cellranger ver.7.0 (10x Genomics). Read count data was merged, principal component analysis (PCA), and uniform Manifold Approximation and Projection (UMAP) were analyzed by Seurat (ver. 4.2.0)[84]. Nuclei with high mitochondrial gene content (>10%) or high unique molecular identifier (UMIs) (>10,000) were excluded. Each dataset was merged by using Canonical correlation analysis. The merged dataset was clustered by Seurat, K-nearest neighbor (KNN) graph, and UMAP. Non-biased clustering was done by using weighted nearest neighbor analysis, and each cluster was annotated based on variable gene expression. Average gene expression and frequency of cells expressing a specific gene were calculated based on cell type or lesion type.

### Reanalysis of single-cell RNA sequencing datasets of CSF cells
We used published datasets of scRNAseq of CSF cells (GSE133028[20], GSE163005[21], GSE138266[22], PRJNA866296[24], GSE172003[23], GSE277954[25]), representing 34 MS and 12 control samples (age: 36.8 ± 10.7 vs. 32.2 ± 7.1 years, respectively). In addition, we analyzed the scRNAseq dataset of CSF cells representing 13 MS participants (5 with RR-MS, 2 with SP-MS, 6 with PP-MS) and 5 controls (1 with non-inflammatory neurological disorder and 4 HC) that has been generated as part of our Natural History cohort and has been published previously, with the raw data deposited under accession number GSE286068[85]. Each raw data was counted, and QC was checked by Cellranger ver.7.0 (10X Genomics, USA). Doublet cells were predicted and excluded by using DoubletFinder V2.0[86], and the protein library was denoised by using dsb package[87]. The cells with high mitochondrial gene (>10%) or high unique molecular identifier (UMIs) (>8000) were excluded from further analysis. Filtered out Read count data was merged by using Canonical correlation analysis (Seurat V4, reciprocal PCA method). The merged dataset was clustered by Seurat, K-nearest neighbor (KNN) graph, and UMAP. Non-biased clustering was done by using weighted nearest neighbor analysis and annotated by using PBMC data[84]. Average gene expression and frequency of cells expressing specific gene were calculated based on cell type.

### Statistics & Reproducibility
We analyzed associations between age/sex-adjusted somamers and various disability and demographic outcomes in untreated MS samples using linear models where somamer was a dependent variable and clinical outcome and sex were independent variables. We tested both the main effect of sex and outcome on somamer levels and their interaction effect in the model. From each model, we extracted coefficients for intercept, outcome, sex, and outcome: sex interaction, along with their respective two-sided *p*-values. We also extracted $R^2$ (proportion of variance explained) with *p*-value of the whole multiple linear regression model that was FDR-adjusted for multiple comparisons (Supplementary data 3). To identify somamers significantly associated with an outcome, we filtered those that passed outcome coefficient *p*-value ≤ 0.05 and the whole model FDR-adjusted *p*-value ≤ 0.05 cutoffs. Somamers affected by sex were identified by further filtering of sex coefficient unadjusted *p*-value and sex: outcome interaction unadjusted *p*-value ≤ 0.05 cutoffs.

We tested differences in somamers levels between two groups of samples (e.g., MS vs HC, Females vs Males, propensity-matched groups) using a two-sided Wilcoxon rank-sum test. The *p*-values were FDR-adjusted for multiple comparisons across the tested somamers.

Correlations between outcome and somamer were generally tested using the Spearman test, resulting in Spearman Rho coefficient and *p*-value, FDR-adjusted for multiple comparisons.

This was a prospectively acquired cohort of samples; no statistical method was used to predetermine sample size.

Two types of outlier analysis were performed: 1. In the cohort of healthy participants that was used to regress out physiological aging and sexual dimorphism effects, we eliminated outliers that fell outside the range of cohort median +/− 1.5 * IQR for each of the measured protein levels (assuming that in a relatively homogeneous population of participants in the absence of neuroimmunulogical disease, the outliers are most likely a result of a technical bias). 2. In the cohort of multiple sclerosis (MS) samples, which were adjusted for healthy aging and sex differences, we identified outliers as values outside of the range of MS cohort median +/− 3 * IQR. These outlier values were then floored to the minimum or maximum of the range of the MS cohort for each particular protein. The purpose of this operation was to eliminate extreme outlier values that would negatively affect downstream linear models, depending on normally distributed datasets. The outlier values were not eliminated completely in the MS cohort under the assumption that, unlike in healthy participants, the heterogeneity of the MS disease processes can result in outlier values, rather than all of them being attributed to technical noise.

This was a prospective study investigating the natural history of multiple sclerosis, with no experimental groups

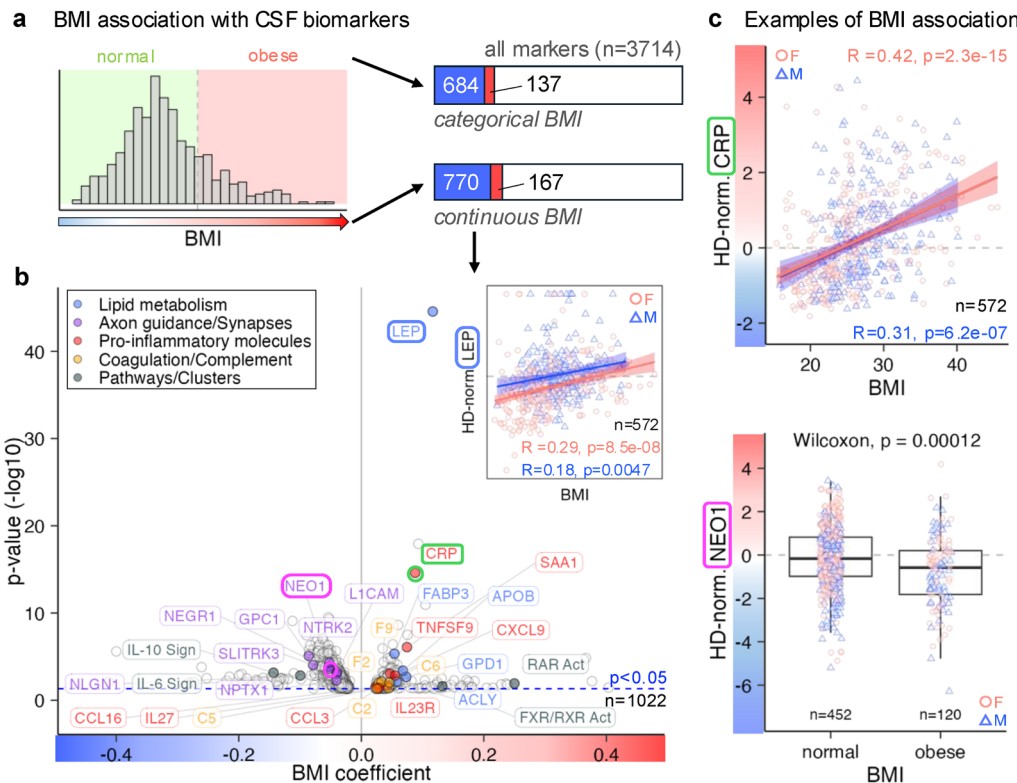

**a** BMI association with CSF biomarkers

**b**

**c** Examples of BMI association

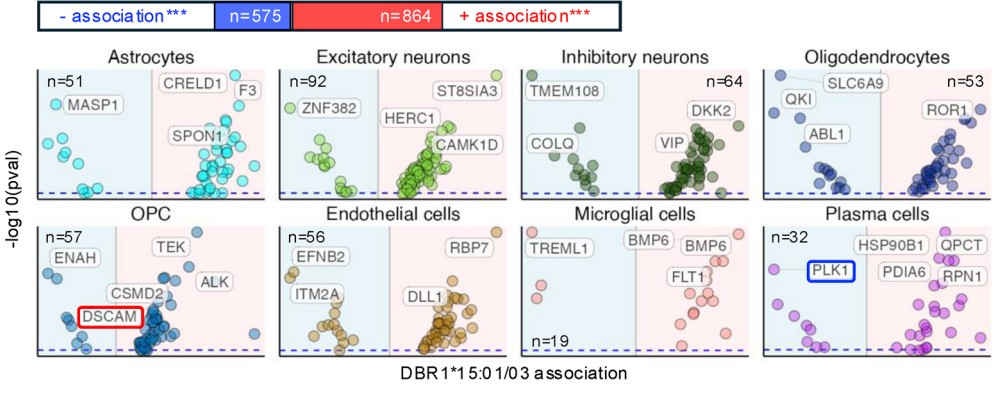

**d** DRB1*15:01/03 haplotype association with CSF biomarkers

all markers (n=3714)

**e** Examples of DRB1*15:01/03 haplotype associations

**f** IPA processes and functions

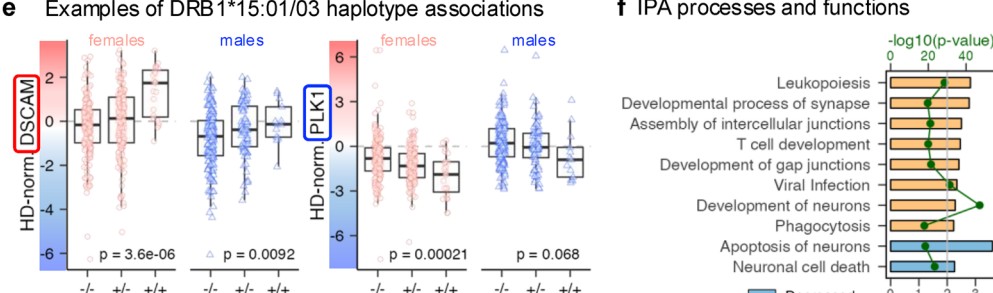

and therefore no randomization. The elastic net model predicting the number of contrast-enhancing lesions, described in this study, was generated in a training cohort of samples and tested in an independent cohort of samples. The randomization of the cohort into training and validation took into account the balance of disease subtypes (relapsing-remitting, primary-, and secondary progressive multiple sclerosis).

Clinical, imaging, and demographic data were collected prospectively before the samples were sent for the proteomic Somascan analysis. The data were QC-ed and locked in the research database. Somascan assay was performed on coded samples by Somalogic personnel blinded to any metadata associated with the samples.

All statistical analyses were performed using RStudio software Version 2023.12.1 + 402 (R version 4.3.3)[88].

**Fig. 9 | Effects of multiple sclerosis (MS) covariates on cerebrospinal fluid (CSF) biomarkers. a** Association of the CSF proteome with body mass index (BMI) was tested using BMI as a continuous and categorical variable (normal <30, obese ≥30), yielding comparable results. **b** Volcano plot of somamer associations with BMI (continuous). Circle colors highlight proteins linked to lipid metabolism (blue), axon guidance/synapses (purple), pro-inflammatory molecules (red), and pathways/clusters (gray). Vertical gray line = coefficient of 0; dashed blue line = unadjusted $p \leq 0.05$. The insert shows the correlation of BMI with leptin (LEP) levels adjusted for healthy control (HC) age and sex. Female samples (pink circles) and male samples (blue triangles) are shown with sex-specific regression lines and 95% confidence intervals (CIs) characterized by Pearson correlation coefficient (R) and unadjusted $p$-value. **c** Examples of BMI-associated somamers. Top: scatter plot of BMI and HC-adjusted C-reactive protein (CRP) levels shows a significant association without sex effect. Bottom: boxplot shows significant differences in NEO1 between normal and obese BMI categories. Females shown as pink circles, males as blue triangles. Significance tested

by two-sided Wilcoxon rank-sum test. Boxplots display medians, quartiles, and whiskers (1.5 × interquartile range [IQR]). Gray dashed lines show HC means. **d** Association of somamers with HLA-DRB1*15:01/03 haplotype tested by regression across three genotypes. One-sided Chi-squared analysis (false discovery rate [FDR]-adjusted) revealed enrichment of astrocyte-, neuronal-, oligodendrocyte-, oligodendroglial precursor cell (OPC)-, endothelial-, microglia-, and plasma cell–specific proteins in homozygous carriers of the susceptibility allele. **e** Examples: DSCAM positively, PLK1 negatively associated with DRB115:01/03 haplotype. Female (pink circles) and male (blue triangles) samples shown; significance tested by two-sided one-way ANOVA. Boxplots as in (**c**), with gray dashed lines marking HC means. **f** Ingenuity Pathway Analysis (IPA) predictions of processes activated (orange) or inhibited (blue) from DRB1*15:01/03-associated proteins. Bar length indicates activation $z$ score; green dots show –log10 $p$-values. All $p$-values of regression coefficients were tested in a two-sided test.

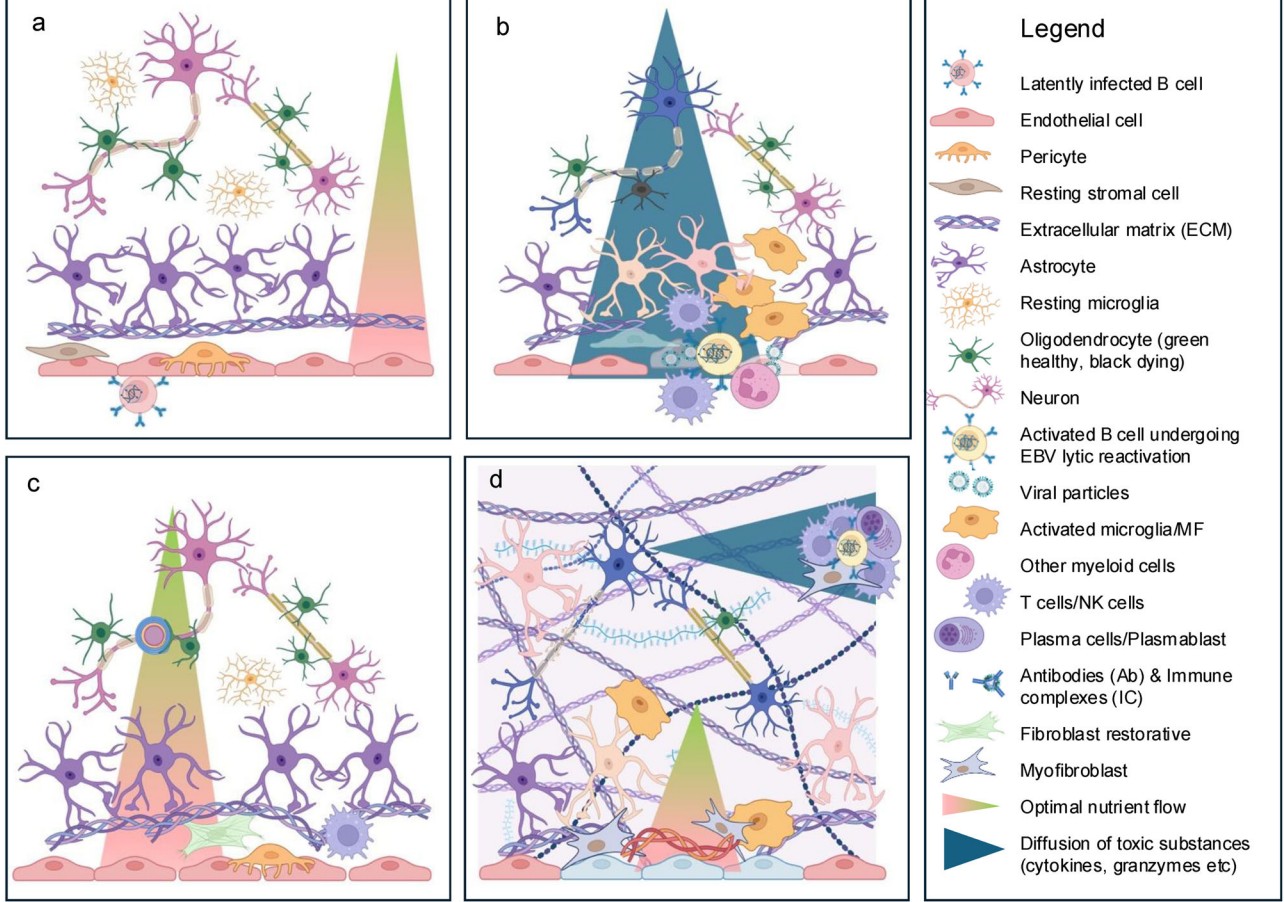

**Fig. 10 | Integrated hypothesis. a** Physiological neurovascular unit and central nervous system (CNS) tissue organization. Under normal conditions, discrete CNS regions receive preferential nutrient flow from adjacent capillary beds. Nutrient exchange occurs mainly at capillaries, whereas immune cell transmigration into CNS tissue typically occurs at post-capillary venules, where slower blood flow facilitates interactions between endothelial adhesion molecules and migrating leukocytes. While the immune cells and mechanisms triggering blood–brain barrier (BBB) opening in multiple sclerosis (MS) remain unclear, evidence suggests Epstein–Barr virus (EBV)-latently infected B cells may occasionally infect CNS endothelial cells during transmigration. The white background represents physiological extracellular matrix (ECM) and neuropil (dense networks of unmyelinated axons, dendrites, synapses, and glial processes). **b** Formation of contrast-enhancing lesions (CELs) and associated CNS injury. If EBV contributes to BBB disruption in MS, it may occur via lytic reactivation in endothelial cells, similar to EBV behavior in nasopharyngeal epithelium. BBB opening initiates ECM degradation, complement and coagulation cascade activation, immune infiltration, and activation of resident CNS cells (astrocytes, microglia, endothelial cells, pericytes, fibroblasts). Inflammatory mediator diffusion (cytokines, granzymes, reactive oxygen/nitrogen

species, serum extravasation) visualized on magnetic resonance imaging (MRI) as CEL following intravenous contrast causes demyelination and axonal damage. Although antiviral CD8+ T cells clear EBV, residual CNS injury remains as non-enhancing T2 lesions, contributing to cumulative T2 lesion load (T2LL). **c** Effective repair response and functional recovery. Recovery depends not only on the initial injury but also on repair quality. Successful repair involves axonal preservation, remyelination, synaptogenesis, and restoration of CNS architecture. Data suggest CNS stromal cells interact with myeloid cells, neurons, and glia to remodel ECM, with stromal cell phenotype determining repair outcome. Restoration of endothelial integrity and neurovascular stoichiometry appears essential for pro-regenerative responses. **d** Failed repair leading to fibrosis, EBV persistence, and neurodegeneration. Maladaptive repair drives stromal cells toward myofibroblast or follicular dendritic cell-like phenotypes, producing non-physiological fibrosis and tertiary lymphoid follicles (TLFs) that harbor EBV-infected B cells. Cycles of reactivation sustain local inflammation. Fibrotic ECM impairs nutrient diffusion (hypoxia), fosters a hostile microenvironment, and drives glial activation, axonal degeneration, and progressive disability accumulation. Created in BioRender. Kosa, P. (2025) https://BioRender.com/9jnhnts.

**Reporting summary**

Further information on research design is available in the Nature Portfolio Reporting Summary linked to this article.

## Data availability

Source data are provided as a Source Data file. Raw data is available in Supplementary Data 1-12. Raw Somascan dataset has been deposited in GitHub (https://github.com/Bielekova-Lab/Bielekova-Lab-Code/blob/master/FormerLabMembers/5k%20study%20R%20codes/input/SOMASCAN_RAW_RFU_5k.csv.zip). RNA-seq datasets analyzed in this study were obtained from publicly available repositories. The datasets are available at the Gene Expression Omnibus (GEO) under accession codes GSE180759[15], GSE118257[16], GSE227781[17] and at the Sequence Read Archive (SRA) under project and PRJNA544731[18,19]. Published datasets of scRNAseq of CSF cells are available at the Gene Expression Omnibus (GEO) under accession codes GSE133028[20], GSE163005[21], GSE138266[22], PRJNA866296[24], GSE172003[23], GSE277954[25], GSE28606[85]. Source data are provided with this paper.

## Code availability

R code used to analyze the data has been deposited in GitHub (https://github.com/Bielekova-Lab/Bielekova-Lab-Code/tree/master/FormerLabMembers/5k%20study%20R%20codes).

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

## Acknowledgements

This research was supported by the Intramural Research Program of the National Institutes of Health (NIH). The contributions of the NIH author(s) were made as part of their official duties as NIH federal employees, are in compliance with agency policy requirements, and are considered Works of the United States Government. However, the findings and conclusions presented in this paper are those of the author(s) and do not necessarily reflect the views of the NIH or the U.S. Department of Health and Human Services. This study was conducted as part of Cooperative Research and Development Agreement between National Institutes of Health, Novartis, and Somalogic.

## Author contributions

B.B. conceived and designed the study and acquired funding; B.B., P.K., curated data, performed formal analyses and interpreted the data, drafted the original manuscript and generated figures; S.A. processed and analyzed RNA seq data; K.L., J.W., C.J.L., performed data analyses; R.M. contributed to sample collection and analysis; Y.K., M.V. acquired and curated brain and spinal cord volumetric data; L.J. provided resources for Somascan analysis; all authors revised the manuscript and approved the submitted version.

## Competing interests

Lori Jennings is an employee of Novartis. No other authors declare any conflict of interest.
