## [Transparent Peer Review file · Nature Communications]

Longitudinal proteomic profiling of cerebrospinal fluid in untreated multiple sclerosis defines evolving disease biology

Corresponding Author: Dr Bibi Bielekova

Version 1:

Reviewer comments:

Reviewer #1

(Remarks to the Author)

I commend the authors for addressing my previous feedback. The revised manuscript is a valuable resource, and I particularly appreciate the improvements in data accessibility i.e. via the Shiny app.

My concerns regarding orthogonal validation have been addressed. While correlated proteins may not directly pertain to the key claims, their inclusion nonetheless increases confidence in the technology, which can still be considered as novel. For a study primarily aimed as a resource, especially where pathways are supported by multiple proteins, this approach is sufficient. The validation in an added cohort of treated individuals is particularly strong; it is noteworthy that treated and untreated patient profiles are so similar. However, as like in my earlier review, I still find the wording regularly quite bold, which is not needed.

Overall, I recommend publication of the revised manuscript, provided the following minor revisions are addressed to further enhance clarity and consistency:

Minor Points

- Please choose a title ideally more informative about the study.
 - Further moderate strength of conclusions, avoiding bold/speculative language.
 - Ensure consistent use of abbreviations and acronyms throughout the manuscript and supplementary materials (e.g. HD vs HV Ext Fig 1a).
 - Row 149: Sentence structure and punctuation require correction for clarity.
 - Rows 221–223: Clarify that “another cohort” refers to a subgroup within the same cohort.
 - Rows 303–305 & Suppl Note 6: Discuss how findings on Schwann cells, typically not CNS cells, can be integrated into picture.
 - Row 450: Typo—correct “matrix-dependend” to “matrix-dependent.”
 - Row 593: Specify that the RNA-seq dataset is of brain tissue.
 - Rows 583–592: Clearly distinguish between modeled and measured CEL#, possibly with distinct abbreviations.
- (line numbers refer to revised manuscript with “simple markup” settings)

(Remarks on code availability)

Reviewer #2

(Remarks to the Author)

The authors have undertaken substantial revisions and provided a detailed rebuttal that effectively addresses the majority of concerns raised in the initial review. I have re-evaluated the manuscript in light of these changes and the point-by-point responses.

Framing of Causality vs. Correlation

The manuscript now consistently frames all associations as hypothesis-generating. The Abstract, Results, and Discussion clearly state that causal inferences cannot be drawn from the data. This resolves my initial concern regarding overstatement of mechanistic conclusions.

Orthogonal and Internal Validation

The authors have added a new Supplementary Note and Extended Figure that present both published and internal orthogonal validation of SomaScan measurements. Importantly, correlations with ELISA assays, NIH clinical laboratory data, and flow-cytometry-based cell counts provide convincing evidence of technical reliability.

Independent Validation Cohort

While no external cohort exists with comparable longitudinal CSF and clinical data, the authors analyzed an independent set of treated MS patients not included in the discovery cohort. Despite expected attenuation, pathway-level associations replicated, with preserved effect sizes and directionality. This constitutes a meaningful internal validation and strengthens confidence in the robustness of the findings.

Batch Effects and Longitudinal Variability

Analyses incorporating CSF storage duration and MRI variability demonstrate negligible impact on outcome modeling. This adequately addresses the concern about potential protocol drift across the 25-year collection period.

Interpretability and Accessibility

The manuscript has been reorganized for clarity, figure-text mismatches corrected, and an interactive public data portal (ShinyApp) created. This resource substantially enhances accessibility and long-term value for the community.

Overreliance on IPA

The authors acknowledge the limitations of pathway inference and mitigate this by integrating scRNA-seq and snRNA-seq data. Their cautious interpretation is now appropriate.

Remaining Caveats

External Validation: The absence of a truly independent cohort remains a limitation. However, I accept the authors' argument that no suitable dataset currently exists and that their internal validation is the most feasible option.

Treatment Effects: Modeling of treatment effects remains limited. The authors acknowledge this openly and frame their analyses as reflective of untreated natural history. This is acceptable within the scope of a resource paper.

(Remarks on code availability)

made.

Response to reviewers:

Reviewer #1 (Remarks to the Author):

I commend the authors for addressing my previous feedback. The revised manuscript is a valuable resource, and I particularly appreciate the improvements in data accessibility i.e. via the Shiny app.

My concerns regarding orthogonal validation have been addressed. While correlated proteins may not directly pertain to the key claims, their inclusion nonetheless increases confidence in the technology, which can still be considered as novel. For a study primarily aimed as a resource, especially where pathways are supported by multiple proteins, this approach is sufficient. The validation in an added cohort of treated individuals is particularly strong; it is noteworthy that treated and untreated patient profiles are so similar. However, as like in my earlier review, I still find the wording regularly quite bold, which is not needed.

Overall, I recommend publication of the revised manuscript, provided the following minor revisions are addressed to further enhance clarity and consistency:

Minor Points

- Please choose a title ideally more informative about the study.
- Further moderate strength of conclusions, avoiding bold/speculative language.

We implemented following additional changes:

- Lines 102-104: Original: "Thus, untreated MS evolution involves ongoing injury to epithelial barriers and inflammation-associated stromal cell-mediated tissue remodeling (see Supplementary Note 1), which facilitates the compartmentalization of inflammation within CNS tissue." Edited: "Thus, CSF biomarkers suggest that untreated MS evolution is characterized by ongoing injury to epithelial barriers and inflammation-associated stromal cell-mediated tissue remodeling (see Supplementary Note 1), which likely facilitates the compartmentalization of inflammation within CNS tissue. "
- Lines 138-139: Original: "Thus, broad activation of anti-viral immunity accompanies CEL formation, linking B cells and macrophages with blood brain barrier (BBB) injury and oligodendrocyte and neuronal loss." Edited: "Thus, CSF biomarkers link CEL formation to proteins associated with blood brain barrier (BBB) injury, intrathecal activation of anti-viral immunity and oligodendroglial and neuronal loss."
- Lines 180-181: Original: "These included glutamatergic synaptic proteins ($p=0.007$), linking T2LL accumulation to synaptic loss." Edited: "These included glutamatergic synaptic proteins ($p=0.007$), implying synaptic loss with T2LL accumulation."
- Lines 186-188: Original: "Thus, T2LL accumulation reflects molecularly chronic inflammatory CNS injury with demyelination and neuronal/synaptic loss. Loss of GABA-ergic signaling likely underlies link between obesity and depression and between depression and MS. ". Edited: "Thus, CSF

biomarkers link T2LL accumulation to chronic inflammatory CNS injury with demyelination and neuronal/synaptic loss. Loss of GABA-ergic signaling may underlie associations of obesity with depression and depression with MS. ”

- Lines 204-206: Original: “ExNeuron proteins were preferentially decreased in people with high BD ($p=0.0128$), indicating that CEL/T2LL-related neuronal loss contributes to cognitive decline.” Edited: “ExNeuron proteins were preferentially decreased in people with high BD ($p=0.0128$), suggesting that CEL/T2LL-related neuronal loss contributes to cognitive decline.”
- Lines 287-289: Original: “Thus, while T2LL-dysproportional brain atrophy and SC injury share proteomic signatures, pwMS with disproportional SC injury exhibit additional features: sustained ICs formation and disrupted IL34/CSF1 signaling axis.” Edited: “Thus, while T2LL-dysproportional brain atrophy and SC injury share proteomic signatures, pwMS with disproportional SC injury have additional CSF proteomic changes suggestive of sustained ICs formation and disrupted IL34/CSF1 signaling axis.”
- Lines 290-291: Original: “Fibrosis-related processes and Schwann cells proteins released during SC injury reflect rates of physical disability progression in longitudinal MS cohort ”; Edited: “Fibrosis-related processes and Schwann cells proteins released during SC injury correlate with rates of physical disability progression in longitudinal MS cohort”
- Lines 350-352: Original: “Third, we identify a critical role for CNS stromal cells in MS evolution. While CEL# declines over time, proteomics reveals ongoing (potentially EBV-mediated) injury to CNS epithelial barriers, driving up fibroblast- and ECM/fibrosis-related proteins in CSF.” Edited: “Third, we identify a critical role for CNS stromal cells in MS evolution. While CEL# declines over time, proteomics reveals ongoing (potentially EBV-mediated) injury to CNS epithelial barriers, and associated increase in fibroblast- and ECM/fibrosis-related proteins in CSF.”

- Ensure consistent use of abbreviations and acronyms throughout the manuscript and supplementary materials (e.g. HD vs HV Ext Fig 1a).

This was corrected – we now consistently use healthy controls (HC).

- Row 149: Sentence structure and punctuation require correction for clarity.

This sentence has been rewritten as 2 sentences: “We restricted the analysis to participants with at least one CEL (i.e., CSF-predicted $CEL\# > 0$). We then quantified CEL-associated axonal injury using residuals from a linear regression of CSF NEFL concentration (y) on CEL count (x), such that higher residuals indicate more destructive lesions (Figure 4i).”

- Rows 221–223: Clarify that “another cohort” refers to a subgroup within the same cohort.

This has been clarified.

- Rows 303–305 & Suppl Note 6: Discuss how findings on Schwann cells, typically not CNS cells, can be integrated into picture.

The Schwann cells are typical cell population in adult human spinal cord. We added this information and citation to spatial transcriptomic study that shows this.

- Row 450: Typo—correct “matrix-depended” to “matrix-dependent.”

Thank you for catching this! Its corrected.

- Row 593: Specify that the RNA-seq dataset is of brain tissue.

This was done.

- Rows 583–592: Clearly distinguish between modeled and measured CEL#, possibly with distinct abbreviations.

This was done.

(line numbers refer to revised manuscript with “simple markup” settings)

Reviewer #2 (Remarks to the Author):

The authors have undertaken substantial revisions and provided a detailed rebuttal that effectively addresses the majority of concerns raised in the initial review. I have re-evaluated the manuscript in light of these changes and the point-by-point responses.

Framing of Causality vs. Correlation

The manuscript now consistently frames all associations as hypothesis-generating. The Abstract, Results, and Discussion clearly state that causal inferences cannot be drawn from the data. This resolves my initial concern regarding overstatement of mechanistic conclusions.

Orthogonal and Internal Validation

The authors have added a new Supplementary Note and Extended Figure that present both published and internal orthogonal validation of SomaScan measurements. Importantly, correlations with ELISA assays, NIH clinical laboratory data, and flow-cytometry-based cell counts provide convincing evidence of technical reliability.

Independent Validation Cohort

While no external cohort exists with comparable longitudinal CSF and clinical data, the authors analyzed an independent set of treated MS patients not included in the discovery cohort. Despite expected attenuation, pathway-level associations replicated,

with preserved effect sizes and directionality. This constitutes a meaningful internal validation and strengthens confidence in the robustness of the findings.

Batch Effects and Longitudinal Variability

Analyses incorporating CSF storage duration and MRI variability demonstrate negligible impact on outcome modeling. This adequately addresses the concern about potential protocol drift across the 25-year collection period.

Interpretability and Accessibility

The manuscript has been reorganized for clarity, figure-text mismatches corrected, and an interactive public data portal (ShinyApp) created. This resource substantially enhances accessibility and long-term value for the community.

Overreliance on IPA

The authors acknowledge the limitations of pathway inference and mitigate this by integrating scRNA-seq and snRNA-seq data. Their cautious interpretation is now appropriate.

Remaining Caveats

External Validation: The absence of a truly independent cohort remains a limitation. However, I accept the authors' argument that no suitable dataset currently exists and that their internal validation is the most feasible option.

Treatment Effects: Modeling of treatment effects remains limited. The authors acknowledge this openly and frame their analyses as reflective of untreated natural history. This is acceptable within the scope of a resource paper.

We would like to thank Reviewers #1 and #2 for their time and effort to thoroughly review our manuscript; their input helped improve the quality of the manuscript greatly.